# Near-Optimal SQ Lower Bounds for Agnostically Learning Halfspaces and ReLUs under Gaussian Marginals

**Ilias Diakonikolas**
University of Wisconsin-Madison
ilias@cs.wisc.edu

**Daniel M. Kane**
University of California, San Diego
dakane@cs.ucsd.edu

**Nikos Zarifis**
University of Wisconsin-Madison
zarifis@wisc.edu

## Abstract

We study the fundamental problems of agnostically learning halfspaces and ReLUs under Gaussian marginals. In the former problem, given labeled examples $(\mathbf{x}, y)$ from an unknown distribution on $\mathbb{R}^d \times \{\pm 1\}$, whose marginal distribution on $\mathbf{x}$ is the standard Gaussian and the labels $y$ can be arbitrary, the goal is to output a hypothesis with 0-1 loss $\mathrm{OPT} + \epsilon$, where $\mathrm{OPT}$ is the 0-1 loss of the best-fitting halfspace. In the latter problem, given labeled examples $(\mathbf{x}, y)$ from an unknown distribution on $\mathbb{R}^d \times \mathbb{R}$, whose marginal distribution on $\mathbf{x}$ is the standard Gaussian and the labels $y$ can be arbitrary, the goal is to output a hypothesis with square loss $\mathrm{OPT} + \epsilon$, where $\mathrm{OPT}$ is the square loss of the best-fitting ReLU. We prove Statistical Query (SQ) lower bounds of $d^{\mathrm{poly}(1/\epsilon)}$ for both of these problems. Our SQ lower bounds provide strong evidence that current upper bounds for these tasks are essentially best possible.

## 1 Introduction

### 1.1 Background and Problem Motivation

We study the fundamental problems of agnostically learning halfspaces and ReLU regression in the distribution-specific agnostic PAC model. In both of these problems, we are given i.i.d. samples from a joint distribution $\mathcal{D}$ on labeled examples $(\mathbf{x}, y)$, where $\mathbf{x} \in \mathbb{R}^d$ is the example and $y \in \mathbb{R}$ is the corresponding label, and the goal is to compute a hypothesis that is competitive with the best-fitting halfspace (with respect to the 0-1 loss) or ReLU (with respect to the square loss) respectively.

A halfspace (or Linear Threshold Function) is any Boolean function $f : \mathbb{R}^d \to \{\pm 1\}$ of the form $f(\mathbf{x}) = \mathrm{sign}\left(\langle \mathbf{w}, \mathbf{x} \rangle + \theta\right)$, where $\mathbf{w} \in \mathbb{R}^d$ is called the weight vector and $\theta$ is called the threshold. (The function $\mathrm{sign} : \mathbb{R} \to \{\pm 1\}$ is defined as $\mathrm{sign}(u) = 1$ if $u \geq 0$ and $\mathrm{sign}(u) = -1$ otherwise.) The task of learning an unknown halfspace from samples is one of the oldest and most well-studied problems in machine learning, starting with the Perceptron algorithm [Ros58] and leading to influential techniques, including SVMs [Vap98] and AdaBoost [FS97]. In the realizable setting [Val84], this learning problem amounts to linear programming and can be solved in polynomial time (see, e.g., [MT94]) without distributional assumptions. In contrast, in the distribution-independent agnostic model [Hau92, KSS94], even *weak* learning is computationally hard [GR06, FGKP06, Dan16].

A line of research [KKMS08, KLS09, ABL17, Dan15, DKS18] has focused on learning halfspaces in the *distribution-specific* agnostic PAC model, where it is assumed that the marginal distribution on the examples is well-behaved. In this paper, we study the important case that the marginal distribution is the standard Gaussian. For concreteness, we formally define this problem.

**Problem 1.1** (Agnostically Learning Halfspaces with Gaussian Marginals)**.** *Let $\mathcal{C}_{\mathrm{LTF}}$ be the class of halfspaces on $\mathbb{R}^d$. Given i.i.d. samples $(\mathbf{x}, y)$ from a distribution $\mathcal{D}$ on $\mathbb{R}^d \times \{\pm 1\}$, where the marginal $\mathcal{D}_{\mathbf{x}}$ on $\mathbb{R}^d$ is the standard Gaussian $\mathcal{N}(\mathbf{0}, \mathbf{I})$ and no assumptions are made on the labels $y$, the goal of the learning algorithm is to output a hypothesis $h : \mathbb{R}^d \to \{\pm 1\}$ such that with high probability we have $\mathbf{Pr}_{(\mathbf{x}, y) \sim \mathcal{D}}[h(\mathbf{x}) \neq y] \leq \mathrm{OPT} + \epsilon$, where $\mathrm{OPT} = \inf_{f \in \mathcal{C}_{\mathrm{LTF}}} \mathbf{Pr}_{(\mathbf{x}, y) \sim \mathcal{D}}[f(\mathbf{x}) \neq y]$.*

The $L_1$-regression algorithm of [KKMS08] solves Problem 1.1 with sample complexity and running time $d^{O(1/\epsilon^2)}$ [DGJ+10, DKN10]. This algorithm is also known to succeed for all log-concave distributions and for certain discrete distributions. A related line of work [ABL17, Dan15, DKS18] has given $\mathrm{poly}(d/\epsilon)$ time algorithms with weaker guarantees, specifically with misclassification error $C \cdot \mathrm{OPT} + \epsilon$, for a universal constant $C > 1$. The fastest known algorithm with optimal error is the one from [KKMS08].

A Rectified Linear Unit (ReLU) is any real-valued function $f : \mathbb{R}^d \to \mathbb{R}_+$ of the form $f(\mathbf{x}) = \mathrm{ReLU}(\langle \mathbf{w}, \mathbf{x} \rangle + \theta)$, where $\mathbf{w} \in \mathbb{R}^d$ is called the weight vector and $\theta$ is called the threshold. (The function $\mathrm{ReLU} : \mathbb{R} \to \mathbb{R}_+$ is defined as $\mathrm{ReLU}(u) = \max\{0, u\}$.) ReLUs are the most commonly used activation functions in modern deep neural networks. Finding the best-fitting ReLU with respect to square-loss is a fundamental primitive in the theory of neural networks. A number of recent works have studied this problem both in terms of finding efficient algorithms and proving hardness results [Sol17, GKKT17, MR18, GKK19, DGK+20]. Similarly to the case of halfspaces, the problem is efficiently solvable in the realizable setting and computationally hard in the distribution-independent agnostic setting [MR18]. Here we study the case of Gaussian marginals, which we now define.

**Problem 1.2** (ReLU Regression with Gaussian Marginals)**.** *Let $\mathcal{C}_{\mathrm{ReLU}}$ be the class of ReLUs on $\mathbb{R}^d$. Given i.i.d. samples $(\mathbf{x}, y)$ from a distribution $\mathcal{D}$ on $\mathbb{R}^d \times \mathbb{R}$, where the marginal $\mathcal{D}_{\mathbf{x}}$ on $\mathbb{R}^d$ is the standard Gaussian $\mathcal{N}(\mathbf{0}, \mathbf{I})$ and no assumptions are made on the labels $y$, the goal of the learning algorithm is to output a hypothesis $h : \mathbb{R}^d \to \mathbb{R}$ such that with high probability we have $\mathbf{E}_{(\mathbf{x}, y) \sim \mathcal{D}}[(h(\mathbf{x}) - y)^2] \leq \mathrm{OPT} + \epsilon$, where $\mathrm{OPT} = \inf_{f \in \mathcal{C}_{\mathrm{ReLU}}} \mathbf{E}_{(\mathbf{x}, y) \sim \mathcal{D}}[(f(\mathbf{x}) - y)^2]$.*

Recent work [DGK+20] gave an algorithm for Problem 1.2 with sample complexity and runtime $d^{\mathrm{poly}(1/\epsilon)}$. While $\mathrm{poly}(d/\epsilon)$ time algorithms are known with weaker guarantees [GKK19, DGK+20], the fastest known algorithm with $\mathrm{OPT} + \epsilon$ error is the one of [DGK+20].

n terms of computational hardness, prior work has given evidence that no $\mathrm{poly}(d, 1/\epsilon)$ time algorithm exists for either problem. Specifically, [KK14] gave a reduction from the problem of learning sparse parities with noise to Problem 1.1. Based on the presumed computational hardness of the former problem, this reduction implies a computational lower bound of $d^{\Omega(\log(1/\epsilon))}$ for Problem 1.1. More recently, [GKK19] gave a qualitatively similar reduction implying a computational lower bound of $d^{\Omega(\log(1/\epsilon))}$ for Problem 1.2. Interestingly, both of these lower bounds cannot be improved in the sense that the corresponding hard instances can be solved in time $d^{O(\log(1/\epsilon))}$.

In summary, the best known algorithms for Problems 1.1 and 1.2 have running time $d^{\mathrm{poly}(1/\epsilon)}$, while the best known computational hardness results give $d^{\Omega(\log(1/\epsilon))}$ lower bounds. This raises the following natural question:

*What is the precise complexity of Problems 1.1 and 1.2?*

Given the lower bounds of [KK14, GKK19], it is conceivable that there exist algorithms for these problems running in time $d^{\mathrm{polylog}(1/\epsilon)}$, i.e., quasi-polynomial in $1/\epsilon$. A positive result of this form would represent significant algorithmic progress in the theory of PAC learning.

*In this paper, we show that the latter possibility is unlikely.* Specifically, we prove Statistical Query (SQ) lower bounds of $d^{\mathrm{poly}(1/\epsilon)}$ for both Problems 1.1 and 1.2. Our SQ lower bounds provide evidence that known algorithms for these problems are essentially best possible.

Before we state our contributions in detail, we give some background on Statistical Query (SQ) algorithms. SQ algorithms are a broad class of algorithms that are only allowed to query expectations

of bounded functions of the distribution rather than directly access samples. The SQ model was introduced by Kearns [Kea98] in the context of supervised learning as a natural restriction of the PAC model [Val84]. Subsequently, the SQ model has been extensively studied in a plethora of contexts (see, e.g., [Fel16] and references therein).

Formally, an SQ algorithm has access to the following oracle.

**Definition 1.3** (STAT Oracle). Let $\mathcal{D}$ be a distribution over some domain $X$ and let $f : X \to [-1, 1]$. A statistical query is a function $q : X \times [-1, 1] \to [-1, 1]$. We define STAT$(\tau)$ to be the oracle that given a query $q(\cdot, \cdot)$ outputs a value $v$ such that $|v - \mathbf{E}_{\mathbf{x} \sim \mathcal{D}}[q(\mathbf{x}, f(\mathbf{x}))]| \leq \tau$, where $\tau > 0$ is the tolerance parameter of the query.

We note that the class of SQ algorithms is rather general and captures most of the known supervised learning algorithms. More broadly, a wide range of known algorithmic techniques in machine learning are known to be implementable using SQs. These include spectral techniques, moment and tensor methods, local search (e.g., Expectation Maximization), and many others (see, e.g., [FGR$^+$13, FGV17]). For the supervised learning problems studied in this paper, all known algorithms with non-trivial performance guarantees are SQ or are easily implementable using SQs.

One can prove lower bounds on the complexity of SQ algorithms via the notion of *Statistical Query (SQ) dimension* [BFJ$^+$94, FGR$^+$13]. A lower bound on the SQ dimension of a learning problem provides an unconditional lower bound on the complexity of any SQ algorithm for the problem.

## 1.2   Our Results and Techniques

We are now ready to formally state our main results. For Problem 1.1 we prove:

**Theorem 1.4.** *Let $d \geq 1$ and $\epsilon \geq d^{-c}$, for some sufficiently small constant $c > 0$. Any SQ algorithm that agnostically learns halfspaces on $\mathbb{R}^d$ under Gaussian marginals within additive error $\epsilon > 0$ requires at least $d^{c/\epsilon}$ many statistical queries to STAT$(d^{-c/\epsilon})$.*

Intuitively, the above statement says that any SQ algorithm for Problem 1.1 requires time at least $d^{\Omega(1/\epsilon)}$. This comes close to the known upper bound of $d^{O(1/\epsilon^2)}$ [KKMS08] and exponentially improves on the best known lower bound of $d^{\Omega(\log(1/\epsilon))}$ [KK14].

For Problem 1.2 we prove:

**Theorem 1.5.** *There exist constants $c, c' > 0$ such that the following holds: For $d \geq 1$ and $\epsilon \geq d^{-c}$, any SQ algorithm with excess error $\epsilon$ for ReLU regression on $\mathbb{R}^d$ under Gaussian marginals requires at least $d^{c/\epsilon^{c'}}$ many statistical queries to STAT$(d^{-c/\epsilon^{c'}})$.*

Intuitively, the above statement says that any SQ algorithm for Problem 1.2 requires time at least $d^{(1/\epsilon)^{\Omega(1)}}$. This qualitatively matches the upper bound of $d^{\mathrm{poly}(1/\epsilon)}$ [DGK$^+$20], up to the degree of the polynomial, and exponentially improves on the best known lower bound of $d^{\Omega(\log(1/\epsilon))}$ [GKK19].

**Discussion.**   The reduction-based hardness of [KK14, GKK19] imply SQ lower bounds of $d^{\Omega(\log(1/\epsilon))}$ for both problems. Our new SQ lower bounds are qualitatively optimal, nearly matching current algorithms. Interestingly, for both problems, our results show a sharp separation in the complexity of obtaining error $O(\mathrm{OPT}) + \epsilon$ (which is $\mathrm{poly}(d/\epsilon)$) versus optimal error $\mathrm{OPT} + \epsilon$. In particular, our lower bounds suggest that the accuracy-runtime tradeoff of known polynomial time approximation schemes (PTAS) for these problems [Dan15, DGK$^+$20] that achieve error $(1 + \gamma)\mathrm{OPT} + \epsilon$, for all $\gamma > 0$, in time $\mathrm{poly}(d^{\mathrm{poly}(1/\gamma)}, 1/\epsilon)$ is qualitatively best possible.

**Technical Overview**   The starting point for our SQ lower bound constructions is the framework of [DKS17]. This work establishes the following: Suppose we have a one-dimensional distribution $A$ that matches its first $k$ moments with the standard one-dimensional Gaussian. Consider the set of distributions $\{\mathbf{P_v}\}$, where $\mathbf{v}$ is any unit vector, such that the projection of $\mathbf{P_v}$ in the $\mathbf{v}$-direction is equal to $A$ and in the orthogonal complement $\mathbf{P_v}$ is an independent standard Gaussian. Then this set of distributions has SQ dimension $d^{\Omega(k)}$. By known results (see, e.g., [Fel16]) this implies that distinguishing such a distribution from the standard Gaussian or learning a distribution with better than $1/\mathrm{poly}(d^k)$ correlation with such a distribution is hard in the SQ model.

To leverage the aforementioned result, a first hurdle that must be overcome is adapting the results of [DKS17] – which apply to the unsupervised task of learning distributions – to the supervised task of learning functions. For Boolean functions $F : \mathbb{R}^d \to \{\pm 1\}$, this is relatively straightforward. Essentially, sampling from the distribution $(\mathbf{x}, F(\mathbf{x}))$ is equivalent to sampling from the conditional distributions of $\mathbf{x}$ conditioned on $F(\mathbf{x}) = 1$ and on $F(\mathbf{x}) = -1$ in a way that is easy to make rigorous in the SQ model. To apply the techniques from [DKS17], we want to construct a one-dimensional Boolean-valued function $f : \mathbb{R} \to \{\pm 1\}$ such that the conditional distributions match moments, or more conveniently so that $\mathbf{E}_{z \sim \mathcal{N}(0,1)}[f(z) z^i] = 0$ for all $0 \leq i \leq k$. Given such a moment matching function, it is not hard to show that the conditional distributions corresponding to the function $F_{\mathbf{v}}(\mathbf{x}) = f(\langle \mathbf{x}, \mathbf{v} \rangle)$, for a unit vector $\mathbf{v}$, are of the type covered by [DKS17].

Of course, for such a construction to have any implications for the problems of agnostically learning halfspaces/ReLUs, we require an additional key property: We want to find a univariate Boolean-valued function $f$ that not only has this kind of matching moments property, but also *correlates non-trivially* with a halfspace or ReLU. If we can guarantee non-trivial correlation, agnostically learning the function $F_{\mathbf{v}}$ with respect to this class will require having a *weak learner* for $F$ (and, for example, preventing the learning from just outputting the identically zero function).

To achieve both aforementioned goals, we use analytic tools to show that there exists an $O(k)$-piecewise constant Boolean-valued function $f$ with $k$ matching moments. Such a function has correlation $\Omega(1/k)$ with some halfspace. For the case or ReLUs, we use an analysis making use of Legendre polynomials to find a function $f' : \mathbb{R} \to [-1, 1]$ with vanishing first $k$ moments and non-trivial correlation with a ReLU. We then show that this can be rounded to a Boolean-valued function with the same guarantees.

**Concurrent Work**   Concurrent and independent work [GGK20] established qualitatively similar SQ lower bounds for agnostically learning halfspaces and ReLUs. Their techniques are different than ours, building on prior SQ lower bounds for learning depth-2 neural networks [DKKZ20, GGJ$^+$], and employing a reduction-based approach.

## 2   Preliminaries

**Notation.** For $n \in \mathbb{Z}_+$, we denote $[n] \stackrel{\text{def}}{=} \{1, \ldots, n\}$ and $\overline{\mathbb{R}} \stackrel{\text{def}}{=} \mathbb{R} \cup \{\pm \infty\}$. For $\mathbf{x} \in \mathbb{R}^d$, and $i \in [d]$, $\mathbf{x}_i$ denotes the $i$-th coordinate of $\mathbf{x}$. We will use $\langle \mathbf{x}, \mathbf{y} \rangle$ for the inner product between $\mathbf{x}, \mathbf{y} \in \mathbb{R}^d$. We will use $\mathbf{E}[X]$ for the expectation of random variable $X$ and $\mathbf{Pr}[\mathcal{E}]$ for the probability of event $\mathcal{E}$.

Let $\mathbf{e}_i$ be the $i$-th standard basis vector in $\mathbb{R}^d$. Let $\mathcal{N}(0, 1)$ denote the standard univariate Gaussian distribution and $\mathcal{N}(\mathbf{0}, \mathbf{I})$ denote the standard multivariate Gaussian distribution. We will use $\phi$ to denote the pdf of the standard Gaussian.

**Correlation and Statistical Query Dimension.** To bound the complexity of SQ learning a concept class $\mathcal{C}$, we will use the standard notion of Statistical Query Dimension [BFJ$^+$94].

For $f, g : \mathbb{R}^d \to \mathbb{R}$, we define the correlation between $f$ and $g$ under the distribution $\mathcal{D}$ to be the expectation $\mathbf{E}_{\mathbf{x} \sim \mathcal{D}}[f(\mathbf{x}) g(\mathbf{x})]$. To prove that the SQ dimension of $\mathcal{C}$ under the distribution $\mathcal{D}$ is large, we need to find a set of functions in the class that are nearly uncorrelated.

**Definition 2.1** (Statistical Query Dimension)**.** For a class of functions $\mathcal{C}$ and distribution $\mathcal{D}$, SQ-DIM$(\mathcal{C}, \mathcal{D}) = s$, if $s$ is the largest integer value for which there exist $s$ functions $f_1, f_2, \ldots, f_s \in \mathcal{C}$ such that for every $i \neq j$, it holds $|\mathbf{E}_{\mathbf{x} \sim \mathcal{D}}[f_i(\mathbf{x}) f_j(\mathbf{x})]| \leq 1/s$.

Our SQ lower bounds will use the following lemma (see, e.g., Theorem 2 of [Szö09]).

**Lemma 2.2.** *Let $\mathcal{C}$ be a concept class of functions on $\mathbb{R}^d$ and $\mathcal{D}$ be a distribution on $\mathbb{R}^d$. Let $s = $ SQ-DIM$(\mathcal{C}, \mathcal{D})$. Any SQ algorithm that outputs a hypothesis with correlation at least $1/s^{1/3}$ from an unknown function in $\mathcal{C}$ requires at least $s^{1/3}/2 - 1$ queries to $\mathrm{STAT}(1/s^{1/3})$.*

We note that the above theorem was initially shown for Boolean-valued functions, but also holds for real-valued functions of bounded norm (see, e.g., [Fel09]).

# 3 SQ Lower Bound for Agnostically Learning Halfspaces

In this section, we prove Theorem 1.4. To do so, we construct a family $\mathcal{F}_k$ of $k$-decision lists of halfspaces on $\mathbb{R}^d$ satisfying the following properties: (1) Any SQ algorithm that weakly learns $\mathcal{F}_k$ requires many high accuracy SQ queries. (2) Each $F \in \mathcal{F}_k$ is non-trivially correlated with a halfspace. Formally, we establish the following statement.

**Proposition 3.1.** *Assuming $d$ is at least a sufficiently large power of $k$, there exists a set $\mathcal{F}_k$ of $k$-decision lists of halfspaces on $\mathbb{R}^d$ such that any SQ algorithm that learns $\mathcal{F}_k$ within 0-1 error $\leq 1/2 - d^{-\Omega(k)}$ with respect to $\mathcal{N}(\mathbf{0}, \mathbf{I})$ needs $d^{\Omega(k)}$ queries to $\mathrm{STAT}(d^{-\Omega(k)})$. Moreover, for any $F \in \mathcal{F}_k$, there is a halfspace $\sigma$ such that $\mathbf{E}_{\mathbf{x} \sim \mathcal{N}(\mathbf{0}, \mathbf{I})}[F(\mathbf{x})\sigma(\mathbf{x})] \geq 1/(2k)$.*

Given the above statement, Theorem 1.4 follows.

*Proof of Theorem 1.4.* Let $\mathcal{A}$ be an agnostic SQ learner for halfspaces under Gaussian marginals. We use $\mathcal{A}$ to weakly learn $\mathcal{F}_k$, for a value of $k$ to be determined. That is, we feed $\mathcal{A}$ a set of i.i.d. labeled examples from an arbitrary function $F \in \mathcal{F}_k$. By definition, algorithm $\mathcal{A}$ computes a hypothesis $h : \mathbb{R}^d \to \{\pm 1\}$ such that $\mathbf{Pr}_{\mathbf{x} \sim \mathcal{N}(\mathbf{0}, \mathbf{I})}[h(\mathbf{x}) \neq F(\mathbf{x})] \leq \mathrm{OPT} + \epsilon$, for $\epsilon > 0$. By Lemma 3.7, it follows that $\mathrm{OPT} \leq 1/2 - 1/(2k)$. Thus, we have that

$$\mathbf{Pr}_{\mathbf{x} \sim \mathcal{N}(\mathbf{0}, \mathbf{I})}[h(\mathbf{x}) \neq F(\mathbf{x})] \leq 1/2 - 1/(2k) + \epsilon .$$

For $\epsilon = 1/(4k)$, Proposition 3.1 gives that $\mathcal{A}$ needs at least $d^{\Omega(1/\epsilon)}$ queries to $\mathrm{STAT}(d^{-\Omega(1/\epsilon)})$. This completes the proof. $\qquad\square$

## 3.1 Proof of Proposition 3.1

The main idea for our construction of a hard-to-learn family of functions $\mathcal{F}_k$ is the following: We first establish the existence of a one-dimensional Boolean-valued function $f : \mathbb{R} \to \{\pm 1\}$ whose first $k$ moments match the first $k$ moments of the standard univariate Gaussian distribution (Proposition 3.2). Importantly, this function $f$ is $(k+1)$-piecewise constant, i.e., there exists a partition of its domain into $k+1$ intervals $I_1, \ldots, I_k$ such that $f$ is constant within each $I_j$. The $k$ points $z_1, z_2, \ldots z_k \in \mathbb{R}$ where the function changes value are called breakpoints.

Given our univariate construction, we construct our family of $d$-dimensional functions by using a copy of this one-dimensional function $f$ oriented in a random direction. More specifically, let $S$ be a set of $2^{d^{\Omega(1)}}$ nearly orthogonal unit vectors on $\mathbb{R}^d$ (Lemma 3.4). Then we define the family $\mathcal{F}_k = \{F_{\mathbf{v}}(\mathbf{x})\}_{\mathbf{v} \in S}$, where $F_{\mathbf{v}}(\mathbf{x}) \stackrel{\text{def}}{=} f(\langle \mathbf{v}, \mathbf{x} \rangle)$ for the univariate function $f$ from Proposition 3.2. Since $f$ is $k$-piecewise constant, each $F \in \mathcal{F}_k$ is a $k$-decision list. From this, it follows that each such $F$ is non-trivially correlated with a halfspace (Lemma 3.7).

Figure 1 shows how the function $F_{\mathbf{v}}$ labels the examples in a 2-dimensional projection.

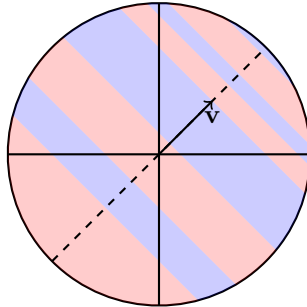

Figure 1: The "red" region is the set of points where $F_{\mathbf{v}}(\mathbf{x}) = -1$ and the "blue" region where $F_{\mathbf{v}}(\mathbf{x}) = 1$.

We show (Proposition 3.1) that any SQ algorithm that can distinguish between an unknown $F_{\mathbf{v}}$ and a function with uniformly random $\pm 1$ labels requires many high accuracy queries.

The key structural result that we require is the following:

**Proposition 3.2.** *For any $k \geq 1$, there exists an at most $(k+1)$-piecewise constant function $f : \mathbb{R} \to \{\pm 1\}$ such that $\mathbf{E}_{z \sim \mathcal{N}(0,1)}[f(z)z^t] = 0$, for every non-negative integer $t < k$.*

Proposition 3.2 is the most technically involved result of this paper. We give its proof in Section 3.2.

In the remainder of this subsection, we prove Proposition 3.1 and Lemma 3.7, assuming Proposition 3.2.

We say that a distribution $A$ has $k$-matching moments with $\mathcal{N}(0,1)$ if $\mathbf{E}_{z \sim A}[z^t] = \mathbf{E}_{\mathbf{x} \sim \mathcal{N}(0,1)}[z^t]$, for all $0 \leq t < k$. Proposition 3.2 implies the following (see Appendix A for the simple proof).

**Lemma 3.3.** *Let $f : \mathbb{R} \to \{\pm 1\}$ be such that $\mathbf{E}_{z \sim \mathcal{N}(0,1)}[f(z)z^t] = 0$, for every non-negative integer $t < k$. For $z \sim \mathcal{N}(0,1)$, define $A \overset{\text{def}}{=} \mathbb{1}\{f(z) = 1\}$ and $B \overset{\text{def}}{=} \mathbb{1}\{f(z) = -1\}$. Then the random variables $A$ and $B$ have $k$-matching moments with $z$.*

We will require the following technical lemmas from [DKS17]. The first lemma says that there exists a large set of unit vectors that are pairwise nearly orthogonal.

**Lemma 3.4** (Lemma 3.7 of [DKS17]). *For any $0 < c < 1/2$, there exists a set $S$ of $2^{\Omega(d^c)}$ unit vectors in $\mathbb{R}^d$ such that for each pair of distinct $\mathbf{u}, \mathbf{v} \in S$, we have $|\langle \mathbf{u}, \mathbf{v} \rangle| \leq O(d^{c-1/2})$.*

If we define the conditional distribution on the event $F_{\mathbf{v}}(\mathbf{x}) = 1$, we can see that the directions orthogonal to $\mathbf{v}$ follow a standard $(d-1)$-dimensional Gaussian distribution. Thus, we define the following distribution for this case.

**Definition 3.5** (High Dimensional Hidden Direction Distribution). *For a univariate distribution $A$ with probability density function $A(z)$ and a unit vector $\mathbf{u} \in \mathbb{R}^d$, consider the distribution over $\mathbb{R}^d$ with pdf $\mathbf{P}_{\mathbf{u}}(\mathbf{x}) = A(\langle \mathbf{u}, \mathbf{x} \rangle) \exp(-\|\mathbf{x} - \langle \mathbf{u}, \mathbf{x} \rangle \mathbf{u}\|_2^2 / 2)/(2\pi)^{(d-1)/2}$. That is, $\mathbf{P}_{\mathbf{u}}$ is the product distribution whose orthogonal projection onto the direction $\mathbf{u}$ is $A$, and onto the subspace perpendicular to $\mathbf{u}$ is the standard $(d-1)$-dimensional Gaussian distribution.*

Let $\mathcal{D}_1, \mathcal{D}_2 : \mathbb{R}^d \to \mathbb{R}_+$ be probability density functions. The $\chi^2$-divergence of $\mathcal{D}_1, \mathcal{D}_2$ is defined as $\chi^2(\mathcal{D}_1, \mathcal{D}_2) \overset{\text{def}}{=} \int_{\mathbb{R}^d} \mathcal{D}_1(\mathbf{x})^2 / \mathcal{D}_2(\mathbf{x}) d\mathbf{x} - 1$. We also define the correlation between $\mathcal{D}_1, \mathcal{D}_2$ and a reference distribution $\mathcal{D}$ as $\chi_{\mathcal{D}}(\mathcal{D}_1, \mathcal{D}_2) \overset{\text{def}}{=} \int_{\mathbb{R}^d} \mathcal{D}_1(\mathbf{x}) \mathcal{D}_2(\mathbf{x}) / \mathcal{D}(\mathbf{x}) d\mathbf{x} - 1$.

The second lemma states that the distributions $\mathbf{P}_{\mathbf{u}}, \mathbf{P}_{\mathbf{v}}$ have correlation depending on the angle of the corresponding vectors.

**Lemma 3.6** (Lemma 3.4 of [DKS17]). *For any $\mathbf{u}, \mathbf{v} \in \mathbb{S}^{d-1}$, let $A : \mathbb{R} \to \mathbb{R}_+$ be the pdf of a distribution that agrees with the first $k$ moments of $\mathcal{N}(0,1)$. Then, we have that*

$$|\chi_{\mathcal{N}(\mathbf{0}, \mathbf{I})}(\mathbf{P}_{\mathbf{u}}, \mathbf{P}_{\mathbf{v}})| \leq |\langle \mathbf{u}, \mathbf{v} \rangle|^{k+1} \chi^2(A, \mathcal{N}(0,1)) .$$

The second statement of Proposition 3.1, establishing non-trivial correlation with a halfspace is shown in the lemma below.

**Lemma 3.7.** *For any $F \in \mathcal{F}_k$, there is a halfspace $\sigma$ such that $\mathbf{E}_{\mathbf{x} \sim \mathcal{N}(\mathbf{0}, \mathbf{I})}[F(\mathbf{x})\sigma(\mathbf{x})] \geq 1/(2k)$.*

The proof of Lemma 3.7 is straightforward and can be found in Appendix A.

We are now ready to prove (the first statement of) Proposition 3.1.

*Proof of Proposition 3.1.* Proposition 3.1] Let $S$ be the set of nearly orthogonal vectors from Lemma 3.4 and $Y$ be a uniform $\pm 1$ random variable. For each $\mathbf{v} \in S$, let $F_{\mathbf{v}}(\mathbf{x}) = f(\langle \mathbf{v}, \mathbf{x} \rangle)$, where $f$ is the function from Proposition 3.2. Let $\mathcal{D}_D$ be the set that contains the distributions $(X, F_{\mathbf{v}}(X))$ for any $\mathbf{v} \in S$ and $X \sim \mathcal{N}(\mathbf{0}, \mathbf{I})$. We will prove that for any $\mathbf{u}, \mathbf{v} \in S$, $\mathbf{u} \neq \mathbf{v}$, we have

$$\chi_{(X,Y)}\left((X, F_{\mathbf{u}}(X)), (X, F_{\mathbf{v}}(X))\right) \leq 2 \cdot |\langle \mathbf{v}, \mathbf{u} \rangle|^{k+1} . \tag{1}$$

To prove Equation (1), for a unit vector $\mathbf{v}$, denote by $A_{\mathbf{v}}$ the conditional distribution of the event $F_{\mathbf{v}} = 1$, and by $B_{\mathbf{v}}$ the conditional distribution of the event $F_{\mathbf{v}} = -1$. Let $\mathcal{D}_{\mathbf{v}}$ be the probability density function of $(X, F_{\mathbf{v}}(X))$. We then have

$$\chi_{(X,Y)}\left((X, F_{\mathbf{u}}), (X, F_{\mathbf{v}})\right) = 2 \int_{\mathbb{R}^d} \frac{\mathcal{D}_{\mathbf{u}}(\mathbf{x}, 1) \mathcal{D}_{\mathbf{v}}(\mathbf{x}, 1)}{\phi(\mathbf{x})} d\mathbf{x} + 2 \int_{\mathbb{R}^d} \frac{\mathcal{D}_{\mathbf{u}}(\mathbf{x}, -1) \mathcal{D}_{\mathbf{v}}(\mathbf{x}, -1)}{\phi(\mathbf{x})} d\mathbf{x} - 1$$

$$= \frac{1}{2} \left(\chi_X(A_{\mathbf{u}}, A_{\mathbf{v}}) + \chi_X(B_{\mathbf{u}}, B_{\mathbf{v}})\right) , \tag{2}$$

where we used that $Y$ gets each label with probability $1/2$ and that $\mathbf{Pr}_{\mathbf{x}\sim\mathcal{N}(\mathbf{0},\mathbf{I})}[F_{\mathbf{v}}(\mathbf{x})=\pm 1]=1/2$. From Lemma 3.6 and Lemma 3.3, we have

$$\left(\chi_X\left(A_{\mathbf{u}},A_{\mathbf{v}}\right)+\chi_X\left(B_{\mathbf{u}},B_{\mathbf{v}}\right)\right)\leq\langle\mathbf{u},\mathbf{v}\rangle^{k+1}\left(\chi^2(A,\mathcal{N}(0,1))+\chi^2(B,\mathcal{N}(0,1))\right). \qquad (3)$$

We also have

$$\chi^2(A,\mathcal{N}(0,1))=\int_{-\infty}^{\infty}A(z)^2/\phi(z)\mathrm{d}z=\int_{-\infty}^{\infty}\frac{\phi(z)^2}{\phi(z)\mathbf{Pr}_{z'\sim\mathcal{N}(0,1)}[f(z')=1]^2}\mathbb{1}\{f(z)=1\}\mathrm{d}z$$

$$=4\int_{-\infty}^{\infty}\phi(z)\mathbb{1}\{f(z)=1\}\mathrm{d}z=2, \qquad (4)$$

where we used the conditional expectation and Equation (5). Putting Equations (2), (3) and (4) together, we get Equation (1). Using Lemma 3.4, we have that $|\langle\mathbf{v},\mathbf{u}\rangle|\leq d^{-(1/2-c)}$, thus

$$\chi_{(X,Y)}\left((X,F_{\mathbf{u}}(X)),(X,F_{\mathbf{v}}(X))\right)\leq\Omega\left(d^{-(k+1)(1/2-c)}\right).$$

To finish our argument, for $\mathbf{u}\neq\mathbf{v}$, we have that

$$\chi_{(X,Y)}\left((X,F_{\mathbf{u}}(X)),(X,F_{\mathbf{v}}(X))\right)=\mathop{\mathbf{E}}_{X\sim\mathcal{N}(\mathbf{0},\mathbf{I})}[F_{\mathbf{u}}(X)F_{\mathbf{v}}(X)]\leq\Omega(d^{-(k+1)(1/2-c)}).$$

Thus, we have that $\mathrm{SQ\text{-}DIM}(\mathcal{F}_k,\mathcal{N}(\mathbf{0},\mathbf{I}))=\min(d^{\Omega(k)},2^{d^c})=d^{\Omega(k)}$, where the last inequality uses the relation between $k$ and $d$. By Lemma 2.2, any SQ algorithm that finds a function $h$ such that $\mathbf{Pr}_{\mathbf{x}\sim\mathcal{N}(\mathbf{0},\mathbf{I})}[F_{\mathbf{v}}(\mathbf{x})\neq h(\mathbf{x})]\leq 1/2-d^{-\Omega(k)}$ needs at least $d^{\Omega(k)}$ queries to $\mathrm{STAT}(d^{-\Omega(k)})$. This completes the proof. $\qquad\square$

## 3.2 Proof of Proposition 3.2

The key lemma for the proof is the following.

**Lemma 3.8.** *Let $m$ and $k$ be positive integers such that $m>k+1$ and $\epsilon>0$. If there exists an $m$-piecewise constant $f:\mathbb{R}\mapsto\{\pm 1\}$ such that $|\mathbf{E}_{z\sim\mathcal{N}(0,1)}[f(z)z^t]|<\epsilon$ for all non-negative integers $t<k$, then there exists an at most $(m-1)$-piecewise constant $g:\mathbb{R}\mapsto\{\pm 1\}$ such that $|\mathbf{E}_{z\sim\mathcal{N}(0,1)}[g(z)z^t]|<\epsilon$ for all non-negative integers $t<k$.*

*Proof.* Let $\{b_1,b_2,\ldots,b_{m-1}\}$ be the breakpoints of $f$. Then let $F(z_1,z_2,\ldots,z_{m-1},z):\overline{\mathbb{R}}^m\mapsto\mathbb{R}$ be an $m$-piecewise constant function with breakpoints on $z_1,\ldots,z_{m-1}$, where $z_1<z_2<\ldots<z_{m-1}$ and $F(b_1,b_2,\ldots,b_{m-1},z)=f(z)$. For simplicity, let $\mathbf{z}=(z_1,\ldots,z_{m-1})$ and define $M_i(\mathbf{z})=\mathbf{E}_{z\sim\mathcal{N}(0,1)}[F(\mathbf{z},z)z^i]$ and let $\mathbf{M}(\mathbf{z})=[M_0(\mathbf{z}),M_1(\mathbf{z}),\ldots M_{k-1}(\mathbf{z})]^T$. It is clear from the definition that $M_i(\mathbf{z})=\sum_{n=0}^{m-1}\int_{z_n}^{z_{n+1}}F(\mathbf{z},z)z^i\phi(z)\mathrm{d}z=\sum_{n=0}^{m-1}a_n\int_{z_n}^{z_{n+1}}z^i\phi(z)\mathrm{d}z$, where $z_0=-\infty$ and $z_m=\infty$ and $a_n$ is the sign of $F(\mathbf{z},z)$ in the interval $(z_n,z_{n+1})$. Note that $a_n=-a_{n+1}$ for every $0\leq n<m$. By taking the derivative of $M_i$ in $z_j$, for $0<j<m$, we get that

$$\frac{\partial}{\partial z_j}M_i(\mathbf{z})=2a_{j-1}z_j^i\phi(z_j)\quad\text{and}\quad\frac{\partial}{\partial z_j}\mathbf{M}(\mathbf{z})=2a_{j-1}\phi(z_j)[1,z_j^1,\ldots,z_j^{k-1}]^T.$$

We now argue that for any $\mathbf{z}$ with distinct coordinates that there exists a vector $\mathbf{u}\in\mathbb{R}^{m-1}$ such that $\mathbf{u}=(\mathbf{u}_1,\ldots,\mathbf{u}_k,0,0,\ldots,0,1)$ and the directional derivative of $\mathbf{M}$ in the $\mathbf{u}$ direction is zero. To prove this, we construct a system of linear equations such that $\nabla_{\mathbf{u}}M_i(\mathbf{z})=0$, for all $0\leq i<k$. Indeed, we have $\sum_{j=1}^k\frac{\partial}{\partial z_j}M_i(\mathbf{z})\mathbf{u}_j=-\frac{\partial}{\partial z_{m-1}}M_i(\mathbf{z})$ or $\sum_{j=1}^k a_{j-1}z_j^i\phi(z_j)\mathbf{u}_j=-a_{m-2}z_{m-1}^i\phi(z_{m-1})$, which is linear in the variables $\mathbf{u}_j$. Let $\hat{\mathbf{u}}$ be the vector with the first $k$ variables and let $\mathbf{w}$ be the vector of the right hand side of the system, i.e., $\mathbf{w}_i=-a_{m-2}z_{m-1}^i\phi(z_{m-1})$. Then this system can be written in matrix form as $\mathbf{V}\mathbf{D}\hat{\mathbf{u}}=\mathbf{w}$, where $\mathbf{V}$ is the Vandermonde matrix, i.e., the matrix that is $\mathbf{V}_{i,j}=\alpha_i^{j-1}$, for some values $\alpha_i$ and $\mathbf{D}$ is a diagonal matrix. In our case, $\mathbf{V}_{i,j}=z_i^{j-1}$ and $\mathbf{D}_{j,j}=2a_{j-1}\phi(z_j)$. It is known that the Vandermonde matrix has full rank iff for all $i\neq j$ we have $\alpha_i\neq\alpha_j$, which holds in our setting. Thus, the matrix $\mathbf{V}\mathbf{D}$ is nonsingular and there exists a solution to the equation. Thus, there exists a vector $\mathbf{u}$ with our desired properties and, moreover, any vector in this direction is a solution of this system of linear equations. Note that the

vector $\mathbf{u}$ depends on the value of $\mathbf{z}$, thus we consider $\mathbf{u}(\mathbf{z})$ be the (continuous) function that returns a vector $\mathbf{u}$ given $\mathbf{z}$.

We define a differential equation for the function $\mathbf{v} : \overline{\mathbb{R}} \mapsto \overline{\mathbb{R}}^{m-1}$, as follows: $\mathbf{v}(0) = \mathbf{b}$, where $\mathbf{b} = (b_1, \ldots, b_{m-1})$, and $\mathbf{v}'(T) = \mathbf{u}(\mathbf{v}(T))$ for all $T \in \mathbb{R}$. If $\mathbf{v}$ is a solution to this differential equation, then we have:

$$\frac{\mathrm{d}}{\mathrm{d}T}\mathbf{M}(\mathbf{v}(T)) = \frac{\mathrm{d}}{\mathrm{d}\mathbf{v}(T)}\mathbf{M}(\mathbf{v}(T))\frac{\mathrm{d}}{\mathrm{d}T}\mathbf{v}(T) = \frac{\mathrm{d}}{\mathrm{d}\mathbf{v}(T)}\mathbf{M}(\mathbf{v}(T))\mathbf{u}(\mathbf{v}(T)) = \mathbf{0} \;,$$

where we used the chain rule and that the directional derivative in $\mathbf{u}(\mathbf{v}(T))$ direction is zero. This means that the function $\mathbf{M}(\mathbf{v}(t))$ is constant, and for all $0 \leq j < k$, we have $|M_j| < \epsilon$, because we have that $|\mathbf{E}_{z \sim \mathcal{N}(0,1)}[F(z_1, \ldots, z_{m-1}, z)z^t]| < \epsilon$. Furthermore, since $\mathbf{u}(\mathbf{v}(T))$ is continuous in $\mathbf{v}(T)$, this differential equation will be well founded and have a solution up until the point where either two of the $z_i$ approach each other or one of the $z_i$ approaches plus or minus infinity (the solution cannot oscillate, since $\mathbf{v}'_{m-1}(T) = 1$ for all $T$).

Running the differential equation until we reach such a limit, we find a limiting value $\mathbf{v}^*$ of $\mathbf{v}(T)$ so that either:

1. There is an $i$ such that $\mathbf{v}_i^* = \mathbf{v}_{i+1}^*$, which gives us a function that is at most $(m-2)$-piecewise constant, i.e., taking $F(\mathbf{v}^*, z)$.

2. Either $\mathbf{v}_{m-1}^* = \infty$ or $\mathbf{v}_1^* = -\infty$, which gives us an at most $(m-1)$-piecewise constant function, i.e., taking $F(\mathbf{v}^*, z)$. Since when the $\mathbf{v}_{m-1}^* = \infty$, the last breakpoint becomes $\infty$, we have one less breakpoint, and if $\mathbf{v}_1^* = -\infty$ we lose the first breakpoint.

Thus, in either case we have a function with at most $m-1$ breakpoints and the same moments. This completes the proof. $\qquad\square$

We also require the following simple fact (see Appendix A for the proof), establishing the existence of a $k'$-piecewise constant Boolean-valued function (for some finite $k'$), satisfying the desired moment conditions.

**Fact 3.9.** *For any $\epsilon > 0$, there exists a $(k/\epsilon)^{O(k)}$-piecewise constant function $f : \mathbb{R} \mapsto \{\pm 1\}$ such that $|\mathbf{E}_{z \sim \mathcal{N}(0,1)}[f(z)z^t]| \leq \epsilon$, for every integer $0 \leq t < k$.*

Proposition 3.2 follows from the above using a compactness argument.

*Proof of Proposition 3.2.* For every $\epsilon > 0$, using the function $f'$ from Fact 3.9 and Lemma 3.8, we can obtain a function $f_\epsilon$ such that $|\mathbf{E}_{z \sim \mathcal{N}(0,1)}[f_\epsilon(z)z^t]| \leq \epsilon$, for every non-negative integer $t < k$ and the function $f_\epsilon$ is at most $(k+1)$-piecewise constant. Let $\mathbf{M} : \overline{\mathbb{R}}^k \mapsto \mathbb{R}^k$, where $M_i(\mathbf{b}) = \sum_{n=0}^{k}(-1)^{n+1}\int_{b_n}^{b_{n+1}} z^i \phi(z)\mathrm{d}z$ and $b_1 \leq b_2 \leq \ldots \leq b_k$, $b_0 = -\infty$ and $b_{k+1} = \infty$. Here we assume without loss of generality that before the first breakpoint the function is negative because we can always set the first breakpoint to be $-\infty$. It is clear that the function $\mathbf{M}$ is a continuous map and $\overline{\mathbb{R}}^{k+1}$ is a compact set, thus $\mathbf{M}\left(\overline{\mathbb{R}}^{k+1}\right)$ is a compact set. We also have that for every $\epsilon > 0$ there is a point $\mathbf{b} \in \overline{\mathbb{R}}^{k+1}$ such that $|\langle \mathbf{M}(\mathbf{b}), \mathbf{e}_i \rangle| \leq \epsilon$. Thus, from compactness, we have that there exists a point $\mathbf{b}^* \in \overline{\mathbb{R}}^{k+1}$ such that $\mathbf{M}(\mathbf{b}^*) = \mathbf{0}$. This completes the proof. $\qquad\square$

## 4  SQ Lower Bound for ReLU Regression

In this section, we give the proof of Theorem 1.5. To prove our theorem, we construct a class $\mathcal{F}$ of Boolean-valued functions that is SQ hard to weakly learn. We use the SQ hardness of $\mathcal{F}$ to show that agnostically learning a ReLU with respect the square loss is also SQ hard.

Our proof depends critically on the following technical result.

**Proposition 4.1.** *For all $k \geq 1$, there exists an $O(k)$-piecewise constant function $f : \mathbb{R} \to \{\pm 1\}$ such that $\mathbf{E}_{z \sim \mathcal{N}(0,1)}[f(z)z^t] = 0$ for every non-negative integer $t \leq k$, and $\mathbf{E}_{z \sim \mathcal{N}(0,1)}[f(z)\mathrm{ReLU}(z)] \geq 1/\mathrm{poly}(k)$.*

To prove Proposition 4.1, we first make essential use of Legendre polynomials to construct an explicit function $f' : \mathbb{R} \to [-1, 1]$ with the correct properties. By rounding $f'$ to a Boolean-valued function, it is not hard to show that for every $\epsilon > 0$ there is a Boolean-valued function whose first $k$ moments are at most $\epsilon$, and whose correlation with a ReLU is at least $1/\text{poly}(k)$. Using a slight variation of the techniques from Lemma 3.8, we can obtain a function with these moments that is $O(k)$-piecewise constant. Taking a limit of such functions with $\epsilon$ tending to 0 gives the proposition.

The hard family of functions to learn will then be $\mathcal{F} = \{F_{\mathbf{v}}\}_{\mathbf{v} \in S}$, where $S$ is a set of nearly orthogonal unit vectors, and $F_{\mathbf{v}}(\mathbf{x}) = C \cdot f(\langle \mathbf{v}, \mathbf{x} \rangle)$, for $C > 0$ some appropriately chosen number of size polynomial in $k$. Since $f(\langle \mathbf{v}, \mathbf{x} \rangle)$ correlates with $\text{ReLU}(\langle \mathbf{v}, \mathbf{x} \rangle)$, taking $C$ sufficiently large, we have

$$\mathbf{E}\left[(F_{\mathbf{v}}(\mathbf{x}) - \text{ReLU}(\langle \mathbf{v}, \mathbf{x} \rangle))^2\right] < \mathbf{E}[F_{\mathbf{v}}(\mathbf{x})^2]\left(1 - 1/\text{poly}(k)\right) \ .$$

If $\epsilon$ is a sufficiently small polynomial in $1/k$, any learner would need to return a function $g$ where $\mathbf{E}[(F_{\mathbf{v}}(\mathbf{x}) - g(\mathbf{x}))^2] < \mathbf{E}[F_{\mathbf{v}}(\mathbf{x})^2](1 - 1/\text{poly}(k))$. This implies both that $\mathbf{E}[(1/C^2)g^2]$ is not too large and that $g$ correlates non-trivially with the Boolean-valued function $(1/C)F_{\mathbf{v}}$. However, since the class of functions $\{(1/C)F_{\mathbf{v}}\}_{\mathbf{v} \in S}$ has large SQ-dimension (because of the moment matching property established by Proposition 4.1), Theorem 1.5 follows from Lemma 2.2.

Due to space limitations, the full proof can be found in Appendix B.

## Broader Impact

Our work fits within the broader agenda of supervised learning in the presence of adversarial label noise. Specifically, we explore the fundamental computational limitations of learning halfspaces with label noise, even under very benign distributional assumptions. It has been a plausible conjecture that strong distributional assumptions make robust learning easy in practice. Understanding to what extent this may be true (or not), can have practical implications in the design of practical learners in such noisy environments.

The primary focus of our work is theoretical and in particular we establish computational lower bounds. As such, we do not expect our results to have immediate societal impact. Nonetheless, we believe that our findings provide useful insights, as they suggest that state-of-the-art algorithms for the problem we study cannot be substantially improved.

## Acknowledgments and Disclosure of Funding

Ilias Diakonikolas is supported by NSF Award CCF-1652862 (CAREER), a Sloan Research Fellowship, and a DARPA Learning with Less Labels (LwLL) grant. Daniel M. Kane is supported by NSF Award CCF-1553288 (CAREER) and a Sloan Research Fellowship. Nikos Zarifis is supported in part by a DARPA Learning with Less Labels (LwLL) grant.

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
