[Supplementary Material]

## Supplementary Material

## A Omitted Proofs from Section 3

### A.1 Proof of Lemma 3.3

We prove the lemma for the random variable $A$. The proof for $B$ is similar. From the definition of $f$, we have that $\mathbf{E}_{z \sim \mathcal{N}(0,1)}[f(z)] = 0$, thus

$$\mathop{\mathbf{E}}_{z \sim \mathcal{N}(0,1)}[\mathbb{1}\{f(z) = 1\}] = \mathop{\mathbf{E}}_{z \sim \mathcal{N}(0,1)}[\mathbb{1}\{f(z) = -1\}]$$

or equivalently

$$\mathop{\mathbf{E}}_{z \sim \mathcal{N}(0,1)}[\mathbb{1}\{f(z) = 1\}] = \frac{1}{2} . \tag{5}$$

Similarly, from $\mathbf{E}_{z \sim \mathcal{N}(0,1)}[f(z)z^t] = 0$, we have

$$\mathop{\mathbf{E}}_{z \sim \mathcal{N}(0,1)}[z^t \mathbb{1}\{f(z) = 1\}] = \mathop{\mathbf{E}}_{z \sim \mathcal{N}(0,1)}[z^t \mathbb{1}\{f(z) = -1\}] . \tag{6}$$

Let $\phi(z|f(z) = 1)$ be the probability distribution of $z$ conditional that $f(z) = 1$. We have that

$$\mathop{\mathbf{E}}_{z \sim A}[z^t] = \int_{-\infty}^{\infty} z^t \phi(z|f(z) = 1) \mathrm{d}z = \int_{-\infty}^{\infty} z^t \frac{\phi(z)}{\mathbf{Pr}_{z' \sim \mathcal{N}(0,1)}[f(z') = 1]} \mathbb{1}\{f(z) = 1\} \mathrm{d}z$$

$$= 2 \int_{-\infty}^{\infty} z^t \phi(z) \mathbb{1}\{f(z) = 1\} \mathrm{d}z = \int_{-\infty}^{\infty} z^t \phi(z) \mathbb{1}\{f(z) = 1\} \mathrm{d}z + \int_{-\infty}^{\infty} z^t \phi(z) \mathbb{1}\{f(z) = -1\} \mathrm{d}z$$

$$= \int_{-\infty}^{\infty} z^t \phi(z) \mathrm{d}z = \mathop{\mathbf{E}}_{z \sim \mathcal{N}(0,1)}[z^t] ,$$

where we used Equations (5), (6).

### A.2 Proof of Lemma 3.7

We start by noting that each $F \in \mathcal{F}_k$ is of the form $F_{\mathbf{v}}(\mathbf{x}) = f(\langle \mathbf{v}, \mathbf{x} \rangle)$, where $f$ is the function from Proposition 3.2 and $\mathbf{v} \in S$. We will take $\sigma$ to be $\sigma(\mathbf{x}) = \text{sign}(\langle \mathbf{v}, \mathbf{x} \rangle + \beta)$. Let $z_1, \ldots, z_k$ be the breakpoints of $f(z)$. We will show that if we set the value of $\beta$ to a breakpoint, then the result follows.

Let $a_{i+1} = \int_{z_i}^{z_{i+1}} \phi(z) \mathrm{d}z$ for $0 < i < k + 1$, $a_1 = \int_{-\infty}^{z_1} \phi(z) \mathrm{d}z$ and $a_{k+1} = \int_{z_k}^{\infty} \phi(z) \mathrm{d}z$. Let $\beta = z_l$, for a breakpoint $z_l$, and $b = \text{sign}(f((z_l + z_{l+1})/2))$. Then we have that

$$\mathop{\mathbf{E}}_{\mathbf{x} \sim \mathcal{N}(\mathbf{0}, \mathbf{I})}[F_{\mathbf{v}}(\mathbf{x})\sigma(\mathbf{x})] = 2 \int_{z_l}^{\infty} f(z)\phi(z) \mathrm{d}z = 2b \sum_{j=l}^{k+1} (-1)^{j-l} a_j ,$$

where the first equality holds because

$$\mathop{\mathbf{E}}_{\mathbf{x} \sim \mathcal{N}(\mathbf{0}, \mathbf{I})}[F_{\mathbf{v}}(\mathbf{x})\mathbb{1}\{\mathbf{x} \in A\}] = - \mathop{\mathbf{E}}_{\mathbf{x} \sim \mathcal{N}(\mathbf{0}, \mathbf{I})}[F_{\mathbf{v}}(\mathbf{x})\mathbb{1}\{\mathbf{x} \in A^c\}] ,$$

for any $A \subseteq \mathbb{R}^d$. From the fact that $\sum_{i=1}^{k+1} a_i = 1$, it follows that there exists an index $i$ such that $a_i \geq 1/(k + 1)$. Assume, for the sake of contradiction, that for all $l > i$ we have that $|\sum_{j=l}^{k+1} (-1)^{l-j} a_j| \leq (1/4k)$, since otherwise there exists a breakpoint that satisfies the equation. Then, for $b = \text{sign}(f((z_i + z_{i+1})/2))$, we have that either $2b(\sum_{j=i+1}^{k+1} (-1)^{j-i} a_j + a_i) \geq (1/2k)$ or $2b(\sum_{j=i+1}^{k+1} (-1)^{j-i} a_j + a_i) \leq -(1/2k)$. In the former case, we are done. In the latter case, the halfspace $-\sigma(\mathbf{x})$ satisfies the desired correlation property.

### A.3 Proof of Fact 3.9

We define $f$ to take alternative values $\pm 1$ in intervals of length $s$. Let us denote $I_i = (is, (i + 1)s)$ for $-1/(s\epsilon^k) \leq i \leq 1/(s\epsilon^k)$ for an integer $i$. If $f(z) = 1$ for $z \in I_i$, then we have $f(z) = -1$ for

$z \in I_{i+1}$. Moreover, we will have that $f(z) = 1$ for $z \leq -1/\epsilon^k$ and $f(z) = -1$ for $z > 1/\epsilon^k$. We assume that the number of constant pieces is even for simplicity. To prove that for all $0 \leq t < k$, $\mathbf{E}_{z \sim \mathcal{N}(0,1)}[f(z)z^t] < 4\epsilon$, observe that for all even moments the expectation is equal to zero. So it suffices to prove the desired statement for odd moments. Note that $\mathbf{E}_{z \sim \mathcal{N}(0,1)}[z^t f(z)\mathbb{1}\{z \geq 0\}] = \mathbf{E}_{z \sim \mathcal{N}(0,1)}[|z^t|f(z)\mathbb{1}\{z < 0\}]$ for odd moments. Thus, we will prove that $\mathbf{E}_{z \sim \mathcal{N}(0,1)}[z^t f(z)\mathbb{1}\{z \geq 0\}] \leq 2\epsilon$. We have that

$$\int_{1/\epsilon^k}^{\infty} z^t \phi(z)\mathrm{d}z \leq \epsilon^{k/t} \ ,$$

where we used the inequality $\mathbf{Pr}_{z \sim \mathcal{N}(0,1)}[|z|^t \geq y] \leq \frac{1}{\sqrt{2\pi}y^{1/t}}e^{-y^{2/t}} \leq 1/y^{1/t}$. Moreover, we bound from above the absolute ratio between two subsequent regions, i.e., $\left| \frac{\mathbf{E}_{z \sim \mathcal{N}(0,1)}[z^t f(z)\mathbb{1}\{z \in I_i\}]}{\mathbf{E}_{z \sim \mathcal{N}(0,1)}[z^t f(z)\mathbb{1}\{z \in I_{i+1}\}]} \right|$. For $i \geq 0$, we have that

$$\frac{\int_{is}^{(i+1)s} z^t \phi(z)\mathrm{d}z}{\int_{(i+1)s}^{(i+2)s} z^t \phi(z)\mathrm{d}z} \leq \frac{s((i+1)s)^t \phi(is)}{s((i+1)s)^t \phi((i+2)s)} = e^{2is^2 + 2s^2} \leq 1 + 3is^2 + 9i^2 s^4 \ , \tag{7}$$

where in the first inequality we used the maximum value and the minimum of the integral, and in the second one we used that $e^x \leq 1 + x + x^2$ for $x \leq 1$, which holds for $s < \epsilon^k$. Thus, for two subsequent intervals we have

$$\int_{is}^{(i+1)s} z^t \phi(z)\mathrm{d}z - \int_{(i+1)s}^{(i+2)s} z^t \phi(z)\mathrm{d}z \leq 4is^2 \int_{(i+1)s}^{(i+2)s} z^t \phi(z)\mathrm{d}z \leq 4ks^2 \int_{(i+1)s}^{(i+2)s} z^t \phi(z)\mathrm{d}z \ .$$

On the other direction, from Equation (7) we have that

$$-\int_{is}^{(i+1)s} z^t \phi(z)\mathrm{d}z + \int_{(i+1)s}^{(i+2)s} z^t \phi(z)\mathrm{d}z \geq -4is^2 \int_{(i+1)s}^{(i+2)s} z^t \phi(z)\mathrm{d}z \geq -4ks^2 \int_{(i+1)s}^{(i+2)s} z^t \phi(z)\mathrm{d}z \ .$$

Thus, we have

$$-4ks^2(t-1)!! \leq \sum_{i=0}^{1/(s\epsilon^k)} (-1)^i \int_{is}^{(i+1)s} z^t \phi(z)\mathrm{d}z \leq 4ks^2 \int_{-\infty}^{\infty} z^t \phi(z) = 4ks^2(t-1)!! \ .$$

Choosing $s = \epsilon^{(k+1)/2}/k^k$ and setting $\epsilon = \epsilon/2$, the proof follows.

## B   Omitted Proofs from Section 4

### B.1   Proof of Proposition 4.1

To prove Proposition 4.1, we first need to prove that there exists a function that has non-trivial correlation with the ReLU and whose first $k$ moments are zero.

We have the following crucial proposition.

**Proposition B.1.** *Let $k$ be a positive integer. There exists a function $f : \mathbb{R} \mapsto [-1, 1]$ such that $\mathbf{E}_{z \sim \mathcal{N}(0,1)}[f(z)z^t] = 0$, for $0 \leq t \leq k$, and $\mathbf{E}_{z \sim \mathcal{N}(0,1)}[f(z)\mathrm{ReLU}(z)] > 1/\mathrm{poly}(k)$.*

The proof of Proposition B.1 requires analytic properties of the Legendre poynomials and is deferred to Section B.2.

In the main part of this subsection, we prove Proposition 4.1, assuming Proposition B.1.

In the following lemma, we show that there exists a piecewise constant Boolean-valued function with near-vanishing moments of degree at most $k$ and non-trivial correlation with the ReLU.

**Lemma B.2.** *For any $\epsilon > 0$ and any non-negative integer $k$, there exists a piecewise constant function $G : \mathbb{R} \mapsto \{\pm 1\}$ such that $|\mathbf{E}_{z \sim \mathcal{N}(0,1)}[G(z)z^t]| \leq \epsilon$, for $0 \leq t \leq k$, and $\mathbf{E}_{z \sim \mathcal{N}(0,1)}[G(z)\mathrm{ReLU}(z)] > 1/\mathrm{poly}(k) + O(\epsilon)$.*

*Proof.* The proof is similar to the proof of Fact 3.9. The main difference here is that we need to construct a function that will also have non-trivial correlation with the ReLU. To do this, we use a probabilistic argument to show that there exists a function that is bounded in the range $[-1, 1]$, that has non trivial correlation, and then we discretize the function as in Fact 3.9. Let $f$ be the function from Proposition B.1. We split the interval $[-1, 1]$ into subintervals of length $\delta$ and we define the random piecewise constant function $G$ in each interval $[z_0, z_0 + \delta]$ as $G(z) = 1$ with probability $(1 + \int_{z_0}^{z_0+\delta} f(z)\phi(z)\mathrm{d}z / \int_{z_0}^{z_0+\delta} \phi(z)\mathrm{d}z)/2$ and $G(z) = -1$ with probability $(1 - \int_{z_0}^{z_0+\delta} f(z)\phi(z)\mathrm{d}z / \int_{z_0}^{z_0+\delta} \phi(z)\mathrm{d}z)/2$. Thus, in each interval, we have $\mathbf{E}[G(z)] = \int_{z_0}^{z_0+\delta} f(z)\phi(z)\mathrm{d}z / \int_{z_0}^{z_0+\delta} \phi(z)\mathrm{d}z$. Then, for any $|z_0| \leq 1 - \delta$, we have that

$$
\begin{aligned}
\mathbf{E}\left[\int_{z_0}^{z_0+\delta} G(z)z^t\phi(z)\mathrm{d}z\right] &= \mathbf{E}\left[\int_{z_0}^{z_0+\delta} G(z)(z_0 + O(\delta))^t\phi(z)\mathrm{d}z\right] = \int_{z_0}^{z_0+\delta} f(z)\phi(z)(z_0 + O(\delta))^t\mathrm{d}z \\
&= \int_{z_0}^{z_0+\delta} f(z)\phi(z)z^t\mathrm{d}z + \int_{z_0}^{z_0+\delta} t \cdot O((|z_0| + \delta)^{t-1}\delta)\mathrm{d}z \\
&= \int_{z_0}^{z_0+\delta} f(z)z^t\phi(z)\mathrm{d}z + t \cdot O\left((|z_0| + \delta)^{t-1}\delta^2\right) ,
\end{aligned}
$$

where we used the Taylor series $z^t = (z_0 + O(\delta))^t + t \cdot O(\delta(|z_0| + \delta)^{t-1})$. Thus, we obtain

$$
\mathbf{E}\left[\int_{-1}^{1} G(z)z^t\phi(z)\mathrm{d}z\right] = \int_{-1}^{1} f(z)z^t\phi(z)\mathrm{d}z + t \cdot O(\delta) = t \cdot O(\delta) ,
$$

where we used that all the moments of $f$ with degree at most $k$ are zero and that $|z_0| + \delta \leq 1$. Moreover, for $0 \leq z_0 \leq 1$, it holds that

$$
\mathbf{E}\left[\int_{z_0}^{z_0+\delta} G(z)\mathrm{ReLU}(z)\phi(z)\mathrm{d}z\right] = \mathbf{E}\left[\int_{z_0}^{z_0+\delta} G(z)z\phi(z)\mathrm{d}z\right] = \int_{z_0}^{z_0+\delta} f(z)\mathrm{ReLU}(z)\phi(z)\mathrm{d}z + t \cdot O\left((|z_0| + \delta)^{t-1}\delta^2\right) ,
$$

where we used the same method as before. Thus, it follows that

$$
\mathbf{E}\left[\int_{0}^{1} G(z)\mathrm{ReLU}(z)\phi(z)\mathrm{d}z\right] = \int_{0}^{1} f(z)\mathrm{ReLU}(z)\phi(z)\mathrm{d}z + t \cdot O(\delta) > 1/\mathrm{poly}(k) + t \cdot O(\delta) .
$$

Define the random variable $X_{i,t} = \int_{i \cdot \delta}^{i \cdot \delta + \delta} G(z)z^t\phi(z)\mathrm{d}z$ and $X_t = \sum_{i=-1/\delta}^{1/\delta} X_{i,t}$. Using Hoeffding bounds, we have that

$$
\mathbf{Pr}[|X_t - \mathbf{E}[X_t]| > \sqrt{\delta}\log(4/(t+1))] \leq 1/(2(t+1)) ,
$$

where we used that $|X_{i,t}| \leq \delta$. By the union bound, we get that there is positive probability that all $X_t$ are within $\pm\sqrt{\delta}\log(4/(t+1))$ from the mean value, and thus, from the probabilistic method there is a function with this property. Furthermore, we round the rest of the values of $G(z)$ as in the proof of Fact 3.9 (because $\mathrm{ReLU}(z) = z$ for $z > 0$). Choosing the correct constant value of $\delta$, the result follows. $\qquad\square$

**Lemma B.3.** *Let $m$ and $k$ be positive integers such that $m > 2k + 5$ and $\epsilon > 0$. If there exists an $m$-piecewise constant function $f : \mathbb{R} \mapsto \{\pm 1\}$ such that $|\mathbf{E}_{z \sim \mathcal{N}(0,1)}[f(z)z^t]| < \epsilon$ for all non-negative integers $t \leq k$, and $\mathbf{E}_{z \sim \mathcal{N}(0,1)}[f(z)\mathrm{ReLU}(z)] > 1/\mathrm{poly}(k) + O(\epsilon)$, then there exists an at most $(2k+5)$-piecewise constant function $g : \mathbb{R} \mapsto \{\pm 1\}$ such that $|\mathbf{E}_{z \sim \mathcal{N}(0,1)}[g(z)z^t]| < \epsilon$ for all non-negative integers $t \leq k$ and $\mathbf{E}_{z \sim \mathcal{N}(0,1)}[g(z)\mathrm{ReLU}(z)] > 1/\mathrm{poly}(k) + O(\epsilon)$.*

*Proof.* This proof is similar to the proof of Lemma 3.8. The only difference is that we have to keep also the correlation with the ReLU constant. For completeness, we provide a full proof.

Let $\{b_1, b_2, \ldots, b_{m-1}\}$ be the breakpoints of $f$. Let $F(z_1, z_2, \ldots, z_{m-1}, z) : \overline{\mathbb{R}}^m \mapsto \mathbb{R}$ be an $m$-piecewise constant function with breakpoints on $z_1, \ldots, z_{m-1}$, where $z_1 < z_2 < \ldots < z_{m-1}$ and $F(b_1, b_2, \ldots, b_{m-1}, z) = f(z)$. For simplicity, let $\mathbf{z} = (z_1, \ldots, z_{m-1})$ and define $M_i(\mathbf{z}) =$

$\mathbf{E}_{z\sim\mathcal{N}(0,1)}[F(\mathbf{z},z)z^i]$, for all $0 \le i \le k$ and $M_c(\mathbf{z}) = \mathbf{E}_{z\sim\mathcal{N}(0,1)}[F(\mathbf{z},z)\mathrm{ReLU}(z)]$. Finally, let $\mathbf{M}(\mathbf{z}) = [M_0(\mathbf{z}), M_1(\mathbf{z}), \dots, M_k(\mathbf{z}), arM_c(\mathbf{z})]^T$. It is clear that

$$M_i(\mathbf{z}) = \sum_{n=0}^{m-1}\int_{z_n}^{z_{n+1}} F(\mathbf{z},z)z^i\phi(z)\mathrm{d}z = \sum_{n=0}^{m-1} a_n \int_{z_n}^{z_{n+1}} z^i\phi(z)\mathrm{d}z\ ,$$

and

$$M_c(\mathbf{z}) = \sum_{n=0}^{m-1}\int_{z_n}^{z_{n+1}} F(\mathbf{z},z)z\mathbb{1}\{z>0\}\phi(z)\mathrm{d}z = \sum_{n=0}^{m-1} a_n \int_{z_n}^{z_{n+1}} z\mathbb{1}\{z>0\}\phi(z)\mathrm{d}z\ ,$$

where $z_0 = -\infty$, $z_m = \infty$, and $a_n$ is the sign of $F(\mathbf{z},z)$ in the interval $(z_n, z_{n+1})$. Note that $a_n = -a_{n+1}$ for every $0 \le n < m$. By taking the derivative of $M_c$ and $M_i$ in $z_j$, for $0 < j < m$, we get that

$$\frac{\partial}{\partial z_j}M_i(\mathbf{z}) = 2a_{j-1}z_j^i\phi(z_j) \quad\text{and}\quad \frac{\partial}{\partial z_j}M_c(\mathbf{z}) = \begin{cases} 2a_{j-1}z_j\phi(z_j), & \text{if } a_j > 0 \\ 0, & \text{if } a_j \le 0 \end{cases}\ .$$

Combining the above, we get

$$\frac{\partial}{\partial z_j}\mathbf{M}(\mathbf{z}) = \begin{cases} 2a_{j-1}\phi(z_j)[1, z_j^1, \dots, z_j^k, z_j]^T, & \text{if } z_j > 0 \\ 2a_{j-1}\phi(z_j)[1, z_j^1, \dots, z_j^k, 0]^T, & \text{if } z_j \le 0\ . \end{cases}$$

We first work with the positive breakpoints. Let $i_0$ be the index of the first positive breakpoint and assume that the positive breakpoints are $m' > k + 2$. We argue that there exists a vector $\mathbf{u} \in \mathbb{R}^{m-1}$ such that $\mathbf{u} = (0, \dots, 0, \mathbf{u}_{i_0+1}, \dots, \mathbf{u}_{i_0+k+2}, 0, 0, \dots, 0, 1)$ and the directional derivative of $\mathbf{M}$ in $\mathbf{u}$ is zero. To prove this, we construct a system of linear equations, such that $\nabla_{\mathbf{u}}M_i(\mathbf{z}) = 0$ for all $0 \le i \le k$ and $\nabla_{\mathbf{u}}M_c(\mathbf{z}) = 0$. Indeed, we have $\sum_{j=1}^{k}\frac{\partial}{\partial z_j}M_i(\mathbf{z})\mathbf{u}_j = -\frac{\partial}{\partial z_{m-1}}M_i(\mathbf{z})$ or $\sum_{j=1}^{k} a_{j-1}z_j^i\phi(z_j)\mathbf{u}_j = -a_{m-2}z_{m-1}^i\phi(z_{m-1})$ and $\sum_{j=1}^{k} a_{j-1}z_j\phi(z_j)\mathbf{u}_j\mathbb{1}\{z_j \ge 0\} = -a_{m-2}z_{m-1}\phi(z_{m-1})\mathbb{1}\{z_{m-1} \ge 0\}$, which is linear in the variables $\mathbf{u}_j$. Note that the last equation is the same equation as the $\nabla_{\mathbf{u}}M_1(\mathbf{z}) = 0$, because we have positive breakpoints only. Let $\hat{\mathbf{u}}$ be the vector with the variables from index $i_0 + 1$ to $i_0 + k + 2$, and let $\mathbf{w}$ be the vector of the right hand side of the system, i.e., $\mathbf{w}_i = -a_{m-2}z_{m-1}^i\phi(z_{m-1})$. Then this system can be written in matrix form as $\mathbf{V}\mathbf{D}\hat{\mathbf{u}} = \mathbf{w}$, where $\mathbf{V}$ is the Vandermonde matrix, i.e., the matrix that is $\mathbf{V}_{i,j} = \alpha_i^{j-1}$, for some values $\alpha_i$ and $\mathbf{D}$ is a diagonal matrix. In our case, $\mathbf{V}_{i,j} = z_i^{j-1}$ and $\mathbf{D}_{j,j} = 2a_{j-1}\phi(z_j)$. It is known that the Vandermonde matrix has full rank iff for all $i \ne j$ we have $\alpha_i \ne \alpha_j$, which holds in our setting. Thus, the matrix $\mathbf{V}\mathbf{D}$ is nonsingular and there exists a solution to the equation. Thus, there exists a vector $\mathbf{u}$ with our desired properties and, moreover, any vector in this direction is a solution to this system of linear equations. Note that the vector $\mathbf{u}$ depends on the value of $\mathbf{z}$, thus we consider $\mathbf{u}(\mathbf{z})$ be the (continuous) function that returns a vector $\mathbf{u}$ given $\mathbf{z}$.

We define a differential equation for the function $\mathbf{v} : \overline{\mathbb{R}} \mapsto \overline{\mathbb{R}}^{m-1}$, as follows: $\mathbf{v}(0) = \mathbf{b}$, where $\mathbf{b} = (b_1, \dots, b_{m-1})$, and $\mathbf{v}'(T) = \mathbf{u}(\mathbf{v}(T))$ for all $T \in \overline{\mathbb{R}}$. If $\mathbf{v}$ is a solution to this differential equation, then we have:

$$\frac{\mathrm{d}}{\mathrm{d}T}\mathbf{M}(\mathbf{v}(T)) = \frac{\mathrm{d}}{\mathrm{d}\mathbf{v}(T)}\mathbf{M}(\mathbf{v}(T))\frac{\mathrm{d}}{\mathrm{d}T}\mathbf{v}(T) = \frac{\mathrm{d}}{\mathrm{d}\mathbf{v}(T)}\mathbf{M}(\mathbf{v}(T))\mathbf{u}(\mathbf{v}(T)) = \mathbf{0}\ ,$$

where we used the chain rule and that the directional derivative in the $\mathbf{u}(\mathbf{v}(T))$ direction is zero. This means that the function $\mathbf{M}(\mathbf{v}(t))$ is constant and, for all $0 \le j < k$ we have $|M_j| < \epsilon$, because we have that $|\mathbf{E}_{z\sim\mathcal{N}(0,1)}[F(z_1, \dots, z_{m-1}, z)z^t]| < \epsilon$. Furthermore, since $\mathbf{u}(\mathbf{v}(T))$ is continuous in $\mathbf{v}(T)$, this differential equation will be well founded and have a solution up until the point where either two of the $z_i$ approach each other or one of the $z_i$ approaches plus or to zero (the solution cannot oscillate, since $\mathbf{v}'_{m-1}(T) = 1$ for all $T$).

Running the differential equation until we reach such a limit, we find a limiting value $\mathbf{v}^*$ of $\mathbf{v}(T)$ so that either:

1. There is an $i$ such that $\mathbf{v}_i^* = \mathbf{v}_{i+1}^*$, which gives us a function that is at most $(m-2)$-piecewise constant, i.e., taking $F(\mathbf{v}^*, z)$.

2. $\mathbf{v}^*_{m-1} = \infty$, which gives us an at most $(m-1)$-piecewise constant function, i.e., taking $F(\mathbf{v}^*, z)$. Since when the $\mathbf{v}^*_{m-1} = \infty$, the last breakpoint becomes $\infty$, we have one less breakpoint.

3. $\mathbf{v}^*_{i_0+1} = 0$, which gives us one less positive breakpoint.

By iterating this method, we can get a function $f'$ that has at most $k + 2$ positive breakpoints. For the negative breakpoints, we work in a similar way, with the only difference that $\frac{\partial}{\partial z_j} M_c(\mathbf{z}) = 0$, for all the negative breakpoints, and that the direction we increase has the form $\mathbf{u} = (-1, \mathbf{u}_1, \ldots, 0, \mathbf{u}_{k+2}, 0, \ldots, 0)$. Thus, we get a function $g$ that has at most $2k + 5$ breakpoints, where we can get an extra breakpoint if 0 is a breakpoint. $\square$

*Proof of Proposition 4.1.* For every $\epsilon > 0$, using the function $f'$ from Lemma B.2 in Lemma B.3, we can obtain a function $f_\epsilon$ such that $|\mathbf{E}_{z\sim\mathcal{N}(0,1)}[f_\epsilon(z)z^t]| \le \epsilon$, for every non-negative integer $t \le k$ and $\mathbf{E}_{z\sim\mathcal{N}(0,1)}[f_\epsilon(z)\mathrm{ReLU}(z)] > 1/\mathrm{poly}(k) + O(\epsilon)$. Moreover, the function $f_\epsilon$ is at most $(2k + 5)$-piecewise constant.

Let $\mathbf{M} : \overline{\mathbb{R}}^{2k+5} \mapsto \mathbb{R}^{k+2}$, where $M_i(\mathbf{b}) = \sum_{n=0}^{2k+5}(-1)^{n+1}\int_{b_n}^{b_{n+1}} z^i\phi(z)\mathrm{d}z$, for $0 \le i < k + 2$, and $M_{k+2}(\mathbf{b}) = \sum_{n=0}^{2k+5}(-1)^{n+1}\int_{b_n}^{b_{n+1}} \mathrm{ReLU}(z)\phi(z)\mathrm{d}z$, where $b_0 \le b_1 \ldots \le b_{2k+5}$, $b_0 = -\infty$ and $b_{2k+6} = \infty$. Here we assume without loss of generality that before the first breakpoint the function is negative, because we can always set the first breakpoint to be $-\infty$. It is clear that the function $\mathbf{M}$ is a continuous map and $\overline{\mathbb{R}}^{2k+5}$ is a compact set, thus $\mathbf{M}\left(\overline{\mathbb{R}}^{2k+5}\right)$ is a compact set. We also have that for every $\epsilon > 0$, there is a point $\mathbf{b} \in \overline{\mathbb{R}}^{2k+5}$ such that $|\langle\mathbf{M}(\mathbf{b}), \mathbf{e}_i\rangle| \le \epsilon$, for $0 \le i < k + 2$, and $\langle\mathbf{M}(\mathbf{b}), \mathbf{e}_{k+2}\rangle > 1/\mathrm{poly}(k) + O(\epsilon)$. Thus, from compactness, we have that there exists a point $\mathbf{b}^* \in \overline{\mathbb{R}}^{2k+5}$ such that $|\langle\mathbf{M}(\mathbf{b}^*), \mathbf{e}_i\rangle| = 0$ for $0 \le i < k+2$, and $\langle\mathbf{M}(\mathbf{b}^*), \mathbf{e}_{k+2}\rangle > 1/\mathrm{poly}(k)$. $\square$

## B.2 Proof of Proposition B.1

Below we state some important properties of the Legendre polynomials that we use in our proofs.

**Fact B.4** ([Sze39])**.** *The Legendre polynomials $P_n(z)$, for $n$ non-negative integer, satisfy the following properties:*

(i) $P_n(z)$ *is a degree-$n$ univariate polynomial, with $P_0(z) = 1$ and $P_1(z) = z$.*

(ii) $\int_{-1}^1 P_i(z)P_j(z)\mathrm{d}z = \delta_{ij}\frac{2}{2i+1}$*, for all $i, j$ non-negative integers (orthogonality).*

(iii) $|P_n(z)| \le 1$*, for all $|z| \le 1$ (bounded).*

(iv) $P'_n(z) = \sum_{t=0}^n \frac{2t+1}{2}P_t(z)$ *(closed form of derivative).*

Using the Legendre polynomials, we can construct a function for which the first $k + 1$ moments are zero and which has non-trivial correlation with the ReLU function.

*Proof of Proposition B.1.* Define $f(z) = c\frac{\mathrm{ReLU}(z)-p(z)}{\phi(z)}\mathbb{1}\{z \in [-1, 1]\}$, for a degree-$k$ polynomial $p(z)$ and a constant $c > 0$. Then, we have

$$\mathop{\mathbf{E}}_{z\sim\mathcal{N}(0,1)}[f(z)z^t] = c\int_{-1}^1(\mathrm{ReLU}(z) - p(z))z^t\mathrm{d}z\ .$$

We want $\mathbf{E}_{z\sim\mathcal{N}(0,1)}[f(z)z^t] = 0$, thus we want to find a polynomial $p(z)$ such that

$$\int_{-1}^1 \mathrm{ReLU}(z)z^t\mathrm{d}z = \int_{-1}^1 p(z)z^t\mathrm{d}z\ . \tag{8}$$

Equation (8) is equivalent to saying that for all $0 \le t < k$, it holds

$$\int_{-1}^1 \mathrm{ReLU}(z)P_t(z)\mathrm{d}z = \int_{-1}^1 p(z)P_t(z)\mathrm{d}z\ , \tag{9}$$

because the Legendre polynomials of degree at most $k$ span the space of polynomials of degree at most $k$. Using Fact B.4 (ii) and a standard computation involving orthogonal polynomials, gives that for $p(z) = \sum_{t=0}^{k} \frac{2t+1}{2} P_t(z) \int_{-1}^{1} \mathrm{ReLU}(z) P_t(z) \mathrm{d}z$, Equation (9) and Equation (8) hold. We want the function $f$ to take values inside the interval $[-1, 1]$. To achieve this, we bound from above the constant $c$. It holds that $\int_{-1}^{1} \mathrm{ReLU}(z) P_t(z) \mathrm{d}z \leq 2$, where we used Fact B.4 (iii) and $|\mathrm{ReLU}(z)| \leq 1$ for $|z| \leq 1$. Moreover, we get that

$$|p(z)| \leq 2 \sum_{t=0}^{k} \frac{2t+1}{2} |P_t(z)| \leq k^2 + 2k \leq 2k^2 \ ,$$

for all $|z| \leq 1$. Thus, it must hold that $c \leq g(1)/(2k^2 + 1)$, and by taking $c = g(1)/(2k^2 + 1)$, we get that $|f(z)| \leq 1$.

Next we prove that $\mathbf{E}_{z \sim \mathcal{N}(0,1)}[f(z) \mathrm{ReLU}(z)] > 1/\mathrm{poly}(k)$. We have that

$$\mathop{\mathbf{E}}_{z \sim \mathcal{N}(0,1)}[f(z) \mathrm{ReLU}(z)] = c \int_{-1}^{1} \mathrm{ReLU}(z)(\mathrm{ReLU}(z) - p(z)) \mathrm{d}z = c \int_{-1}^{1} (\mathrm{ReLU}(z) - p(z))^2 \mathrm{d}z \ ,$$

where we used that $\int_{-1}^{1} q(z)(\mathrm{ReLU}(z) - p(z)) \mathrm{d}z = 0$, for any polynomial $q$ of degree at most $k$, and thus it holds for $q(z) = p(z)$. Note that $|p'(z)| \leq 5k^4$ and $|p''(z)| \leq 7k^6 =: N$, because from Fact B.4 (iv), we have that $|P_n'(z)| \leq 2n^2$ and $|P_n''(z)| \leq 4n^4$, for all $|z| \leq 1$. For $\epsilon > 0$ sufficiently small, we then have

$$\int_{-1}^{1} (\mathrm{ReLU}(z) - p(z))^2 \mathrm{d}z \geq \int_{-\epsilon}^{\epsilon} (\mathrm{ReLU}(z) - p(z))^2 \mathrm{d}z \ .$$

Using the Taylor expansion of $p$, we get that there exists a linear function $L$, such that $p(z) = L(z) + O(N\epsilon^2)$, for $|z| \leq \epsilon$. We thus have that

$$\int_{-\epsilon}^{\epsilon} (\mathrm{ReLU}(z) - p(z))^2 \mathrm{d}z = \int_{-\epsilon}^{\epsilon} (\mathrm{ReLU}(z) - L(z) + O(N\epsilon^2))^2 \mathrm{d}z \ .$$

Note that every function can be written as $G(z) = G_{\mathrm{even}}(z) + G_{\mathrm{odd}}(z)$, where $G_{\mathrm{even}}(z)$ is the even part of $G$ and $G_{\mathrm{odd}}(z)$ is the odd part. For $\ell > 0$, it holds that

$$\int_{-\ell}^{\ell} G^2(z) \mathrm{d}z = \int_{-\ell}^{\ell} \left( G_{\mathrm{even}}^2(z) + G_{\mathrm{odd}}^2(z) + 2 G_{\mathrm{even}}(z) G_{\mathrm{odd}}(z) \right) \mathrm{d}z \geq \int_{-\ell}^{\ell} G_{\mathrm{even}}^2(z) \mathrm{d}z \ ,$$

where we used that $\int_{-\ell}^{\ell} G_{\mathrm{even}}(z) G_{\mathrm{odd}}(z) = 0$. Using that $\mathrm{ReLU}(z) = |z|/2 + z/2$, it holds

$$\int_{-\epsilon}^{\epsilon} (\mathrm{ReLU}(z) - L(z) + O(N\epsilon^2))^2 \mathrm{d}z \geq \int_{-\epsilon}^{\epsilon} (|z|/2 - L(0) + O(N\epsilon^2))^2 \mathrm{d}z \ ,$$

where we used that $L$ is linear, thus the even part is $L(0)$. Choosing $\epsilon$ such that $N < \epsilon^{-1}/C$ for a large enough $C > 0$, we have that $||z|/2 - L(0)| \geq \epsilon/8$ for at least half of the interval $[-\epsilon, \epsilon]$. To prove this, note that we have two cases. First, if $L(0) > \epsilon/2$ or $L(0) \leq 0$, this holds trivially. Again in the other case trivially in half the points we have $||z|/2 - L(0)| \geq \epsilon/4$. Moreover, from the choice of $\epsilon$, we have that $N\epsilon^2 \leq \epsilon/C$, thus $||z|/2 - L(0) + O(N\epsilon^2)| \geq |||z|/2 - L(0)| - |O(N\epsilon^2)|| \geq \epsilon/8$ for at least half of the interval. Therefore, we have

$$\int_{-\epsilon}^{\epsilon} (|z|/2 - L(0) + O(N\epsilon^2))^2 \mathrm{d}z \geq \Omega(\epsilon^3) \ .$$

By our choice of $\epsilon$, we have

$$c \int_{-1}^{1} (\mathrm{ReLU}(z) - p(z))^2 \mathrm{d}z \geq c \cdot \Omega(\epsilon^3) \geq c \cdot \Omega(N^{-3}) \geq \Omega(1/k^{20}) \ .$$

This completes the proof. $\qquad \square$

## B.3 Proof of Theorem 1.5

The proof follows using the same construction as in Theorem 1.4, but using the $O(k)$-piecewise constant function $f$ from Proposition 4.1. Let $C(k)$ be a constant that depends on $k$ and $\mathcal{F}_k$ be the family of $O(k)$-decision lists of halfspaces, where each $F_{\mathbf{v}} \in \mathcal{F}_k$ has the form $F_{\mathbf{v}}(\mathbf{x}) = C(k) \cdot f(\langle \mathbf{v}, \mathbf{x} \rangle)$, for a unit vector $\mathbf{v} \in S$, where we use the set $S$ from Lemma 3.4. Let $\mathcal{A}$ be an agnostic SQ learner for ReLUs under Gaussian marginals. We feed $\mathcal{A}$ a set of i.i.d. labeled examples from an arbitrary function $F_{\mathbf{v}} \in \mathcal{F}_k$. By definition, algorithm $\mathcal{A}$ computes a hypothesis $h : \mathbb{R}^d \mapsto \mathbb{R}$ such that

$$\mathop{\mathbf{E}}_{\mathbf{x} \sim \mathcal{N}(\mathbf{0},\mathbf{I})} [(h(\mathbf{x}) - F_{\mathbf{v}}(\mathbf{x}))^2] \leq \inf_{f \in \mathcal{C}_{\mathrm{ReLU}}} \mathop{\mathbf{E}}_{\mathbf{x} \sim \mathcal{N}(\mathbf{0},\mathbf{I})} [(f(\mathbf{x}) - F_{\mathbf{v}}(\mathbf{x}))^2] + \epsilon \ .$$

We denote $\|g\|_2^2 = \mathbf{E}_{\mathbf{x} \sim \mathcal{N}(\mathbf{0},\mathbf{I})}[g(\mathbf{x})^2]$ for a function $g : \mathbb{R}^d \mapsto \mathbb{R}$. Let $C(k) = \frac{\|\mathrm{ReLU}\|_2^2}{\mathbf{E}_{\mathbf{x} \sim \mathcal{N}(\mathbf{0},\mathbf{I})}[f(\langle \mathbf{x}, \mathbf{v} \rangle)\mathrm{ReLU}(\langle \mathbf{x}, \mathbf{v} \rangle)]}$. Then we have that

$$\mathop{\mathbf{E}}_{\mathbf{x} \sim \mathcal{N}(\mathbf{0},\mathbf{I})} [(\mathrm{ReLU}(\langle \mathbf{x}, \mathbf{v} \rangle) - F_{\mathbf{v}}(\mathbf{x}))^2] = \|F_{\mathbf{v}}\|_2^2 + \|\mathrm{ReLU}\|_2^2 - 2 \mathop{\mathbf{E}}_{\mathbf{x} \sim \mathcal{N}(\mathbf{0},\mathbf{I})} [F_{\mathbf{v}}(\mathbf{x})\mathrm{ReLU}(\langle \mathbf{x}, \mathbf{v} \rangle)]$$

$$= C^2(k) \|f\|_2^2 - \|\mathrm{ReLU}\|_2^2 \ .$$

Furthermore, using that $\|f\|_2^2 = 1$ and $\|\mathrm{ReLU}\|_2^2 = 1/2$, if we choose $\epsilon = o(1/C^2(k))$, the algorithm returns a hypothesis such that

$$\mathop{\mathbf{E}}_{\mathbf{x} \sim \mathcal{N}(\mathbf{0},\mathbf{I})} [(h(\mathbf{x}) - F_{\mathbf{v}}(\mathbf{x}))^2] \leq C^2(k) \left(1 - \Omega(1/C^2(k))\right) \ .$$

Thus, from the triangle inequality, we have that $\|h/C(k)\|_2^2 \leq 2 \|f\|_2^2$, and also

$$2 \mathop{\mathbf{E}}_{\mathbf{x} \sim \mathcal{N}(\mathbf{0},\mathbf{I})} \left[ \frac{h(\mathbf{x})}{C(k)} \frac{F_{\mathbf{v}}(\mathbf{x})}{C(k)} \right] \geq \Omega(1/C^2(k)) + \|h\|_2^2 / C^2(k) \geq \Omega(1/C^2(k)) \ .$$

Finally,

$$\mathop{\mathbf{E}}_{\mathbf{x} \sim \mathcal{N}(\mathbf{0},\mathbf{I})} \left[ \frac{h(\mathbf{x})}{\|h\|_2} \frac{F_{\mathbf{v}}(\mathbf{x})}{\|F_{\mathbf{v}}\|_2} \right] \geq \frac{1}{2} \mathop{\mathbf{E}}_{\mathbf{x} \sim \mathcal{N}(\mathbf{0},\mathbf{I})} \left[ \frac{h(\mathbf{x})}{C(k)} \frac{F_{\mathbf{v}}(\mathbf{x})}{C(k)} \right] \geq \Omega(1/C^2(k)) \ .$$

Let $h^*(\mathbf{x}) = \frac{h(\mathbf{x})}{\|h\|_2}$ and $F_{\mathbf{v}}^*(\mathbf{x}) = \frac{F_{\mathbf{v}}(\mathbf{x})}{\|F_{\mathbf{v}}\|_2}$. Then we have that $\mathbf{E}_{\mathbf{x} \sim \mathcal{N}(0,I)} [h^*(\mathbf{x}) F_{\mathbf{v}}^*(\mathbf{x})] \geq \Omega(1/C^2(k))$. Thus, using Proposition 4.1 to bound $C(k)$, we get that

$$\mathop{\mathbf{E}}_{\mathbf{x} \sim \mathcal{N}(\mathbf{0},\mathbf{I})} [h^*(\mathbf{x}) F_{\mathbf{v}}^*(\mathbf{x})] \geq \Omega(1/\mathrm{poly}(k)) \ .$$

Since the function $F_{\mathbf{v}}$ is an $O(k)$-decision list of halfspaces, we can apply Proposition 3.1 to get that any SQ algorithm needs $d^{\Omega(k)}$ queries to $\mathrm{STAT}(d^{-\Omega(k)})$ to get $\mathbf{E}_{\mathbf{x} \sim \mathcal{N}(\mathbf{0},\mathbf{I})} [h^*(\mathbf{x}) F_{\mathbf{v}}^*(\mathbf{x})] \geq d^{-\Omega(k)}$. Thus, in order to learn with error $\mathrm{OPT} + \epsilon$, for $\epsilon = o(1/\mathrm{poly}(k))$, the algorithm $\mathcal{A}$ needs to use $d^{\Omega((1/\epsilon)^c)}$ queries to $\mathrm{STAT}(d^{-\Omega((1/\epsilon)^c)})$, for a constant $c > 0$.