[Reviews · NeurIPS 2020]

Review 1

Summary and Contributions: This paper studies two fundamental problems in learning theory, namely that of *agnostically* learning (i) halfspaces (w.r.t. 0-1 loss) and (ii) single ReLU neurons (w.r.t. squared loss). The paper identifies a key barrier that provides strong evidence that efficient learning algorithms for these problems may not exist, even in the arguably "simple" setting of Gaussian marginals. In particular, the paper considers the "statistical query (SQ) model", which is very well studied in literature and most of the learning algorithms studied in literature can be implemented in the SQ model as well (with few notable exceptions of course). The paper shows that *agnostically* learning halfspaces / ReLUs to error OPT + eps, requires super-polynomial (i.e. d^{poly(1/eps)}) many SQ queries, which improves significantly upon previous lower bounds of d^{polylog(1/eps)}. Moreover, this lower bound is nearly optimal as it matches known upper bounds upto poly factors in the exponent. RESPONSE TO AUTHORS: Thank you for incorporating the comments. My rating for the paper is unchanged, that is, I continue to feel the paper should be accepted.

Strengths: The paper establishes very basic and fundamental results in learning theory. The key takeaway is that one cannot hope for a learning algorithm that achieves a guarantee of OPT + eps in less than d^{poly(1/eps)} run-time, at least in the SQ model. This explains why other prior work had to relax the guarantee to c.OPT + eps to get poly(d/eps) run-time. This will also inform future research on learning algorithms for these problems.

Weaknesses: I don't have anything that I would necessarily call a "weakness". The result would be stronger if the hard family constructed were more "flexible"; the current hard family of functions is such that OPT ~ 1/2-2eps, and it is shown to be hard to learn it to error 1/2-eps. Could there be a function family where OPT is significantly smaller than 1/2 (e.g. OPT = 1/3) and still is hard to learn to error OPT + eps? Or even more strongly, could there be a hard function family for any given choice of OPT > 0?

Correctness: The proofs seem correct to me at a high level.

Clarity: The paper is well written and the proofs are modular and clear.

Relation to Prior Work: Relation to prior work is clearly discussed. The paper does a good job of summarizing a lot of the recent developments related to the problem of learning halfspaces and ReLUs.

Reproducibility: Yes

Additional Feedback: Some technical comments: - Line 124 : I didn't get what is meant by "set of distributions has SQ dimension d^{Omega(k)}". I understand this is not critical to understanding the rest of the paper, but it might be good to quickly define what this means (or cite a definition number in prior work). - Line 169 (Lemma 2.2) : I was surprised to see that the lower bounds against ReLUs are obtained for the SQ model and not the weaker CSQ model (If I understand correctly, in the case of real valued targets, the lower bound we get from SQ-dimension applies only to the CSQ model. Isn't it?). However, after going through the proof of Thm 1.5, I realized that the final lower bound for ReLUs applies also for the SQ model because the target is +/- 1 valued. I think it would be good to clarify this. - Line 228-229: While I technically understand the definition of "correlation" and also follow how it is used in the proofs, it will be good to have some intuition for this definition. I suppose if D_1 = D_2 then it is same as chi^2(D_1, D). Btw, I think chi^2(.,.) is used in Lemma 3.7 without defining it anywhere.


Review 2

Summary and Contributions: This paper studies two very elementary problems in supervised learning: agnostically learning halfspaces and ReLUs. (This review will focus just on the case of halfspaces, which is the more classic learning problem; the results and ideas for ReLUs are qualitatively similar.) The goal in agnosticly (improperly) learning halfspaces is to take samples $(x,y)$ from some distribution on $R^d \times \{+1,-1\}$ and find a hypothesis $h \, : \, R^d \rightarrow \{+1,-1\}$ such that the 0/1 loss of $h$ -- that is, $Pr(h(x) \neq y)$ is as small as possible, ideally $OPT + \epsilon$, where $OPT$ is the least 0/1 loss achieved by any function of the form $sign(\langle{v,x} + \theta)$ for some vector $v$ and constant $\theta$. This is a particularly challenging version of the learning half spaces problem, and it is known to be computationally hard (via e.g. reductions from random xor) without some additional assumptions. A standard such assumption, which this paper focuses on, is that $x$ is Gaussian $N(0,I)$ -- this is the "Gaussian Marginals" in the title. Under this assumption, algorithms which are somewhat computationally efficient are known -- requiring time and samples $d^{1/\epsilon^2}$. (By contrast, if willing to run in exponential time, then $\poly(d,\epsilon)$ samples are enough.) The main question is whether $d^{1/\epsilon^2}$ is the right running time / sample complexity, or is a more efficient algorithm possible? Two significant pieces of evidence suggest it could be possible: -- $\poly(d,\epsilon)$ time and samples algorithms are known with the weaker error guarantee of $O(OPT) + \epsilon$ -- the strongest known lower bounds (proved via reduction from standard complexity assumptions) only rule out algorithm faster than $d^{\log(1/\epsilon)}$. This paper proves a lower bound in the restricted "statistical query" (SQ) model bound which strongly suggests that at least $d^{1/\epsilon}$ samples and time are necessary. RESPONSE TO AUTHORS: Thank you for the clarifications! I continue to feel the paper should be accepted.

Strengths: Essentially resolves the important question: can half spaces be learned agnostically w.r.t. Gaussian marginals in time less than $d^{\poly(1/\epsilon)}$? Prior to this work it seemed wide open whether the right complexity for this fundamental problem should be $d^{\poly(1/\epsilon)}$ or as much as exponentially less than this. While SQ is a restricted model of computation, SQ lower bounds are generally quite believable for this kind of noisy learning problem -- I would view this as a strong no-go result for polynomial or quasi-polynomial time algorithms for this problem. The technical tools to prove the SQ lower bound draw heavily on work of Diakonikolas-Kane-Stewart from FOCS 2017. Once the suggested hard instances are described in the paper it is not too surprising, in light of these FOCS '17 results, that the SQ lower bound arguments can go through -- still, I think there is more than enough technical innovation for a NeurIPS accept. (And constructing the hard instances requires some cleverness!) Learning halfspaces is perhaps the most basic problem in learning theory, and Gaussian marginals are a very natural and well studied assumption. (E.g. the thoroughly-studied random-design linear regression setting is basically the "realizable" case of this problem.) The results are of clear interest to the NeurIPS community.

Weaknesses: There is still a gap between the lower bound of $d^{1/\epsilon}$ and the known running time of $d^{1/\epsilon^2}$. (This is not really a weakness, per se -- it just means that the paper is not quite the final word on the subject.) Because of this gap, though, I think a few things are slightly overclaimed. For instance, the title is a little too strong: "Near-Optimal" should probably be reserved for a lower bound like $d^{1/\epsilon^2 (\log 1/\epsilon)^{O(1)}}$.

Correctness: I did not check all the proofs line-by-line, but once the hard instances are described it seems clear given the FOCS '17 results that the SQ lower bounds should work out.

Clarity: Yes, the paper is well written. (See a few small complaints below.)

Relation to Prior Work: The prior work discussion is clear and comprehensive.

Reproducibility: Yes

Additional Feedback: -- l.42 "optimal error" should be more precise; say OPT+\epsilon -- l.118 "best possible" seems too strong here. couldn't you hope for e.g. $\poly(d^{\log(1/\gamma)}, 1/\epsilon)$? -- l.171 should define $k$-decision list -- proof of proposition 3.3: it would be good to offer some intuition about what is going on here; this is a kind of technical proof. Also, the key point about the $\alpha_i$'s being distinct is kind of buried and it isn't really made clear that this is because the breakpoints $b_i$ are distinct (at least, I think that's why this is true...)


Review 3

Summary and Contributions: Ever since the L1 regression-based algorithm of Kalai-Klivans-Mansour-Servedio '08 for agnostically learning halfspaces over Gaussians to OPT + eps, an outstanding open question has been whether their d^{poly(1/eps)} runtime could be improved. Kothari-Klivans '14 showed a lower bound of d^{log 1/eps} based on noisy parity. The present paper closes this long-standing gap by giving a d^{poly(1/eps)} lower bound (for statistical query algorithms). They also show a similar lower bound for agnostically learning a ReLU over Gaussians to OPT + eps, for which Goel-Karmalkar-Klivans '19 had previously shown a d^{log 1/eps} lower bound, also based on sparse parity. The present paper's techniques use the framework from Diakonikolas-Kane-Stewart '17 for showing SQ lower bounds via embedding low-dimensional moment-matching distributions. The key technical contribution is to design such a moment-matching distribution for halfspaces/ReLUs, e.g. for halfspaces they show there exists a {\pm 1}-valued k-wise-constant function f over the real line for which E_{g~N(0,1)}[f(g)g^t] = 0 for all t < k-1, for any k. The argument is very clever: 1) for any eps, there is a fine enough discretization of a subset of the real line such that the function f which oscillates some f(k,eps) times between \pm 1 over this discretization will satisfy E_{g~N(0,1)}[f(g)g^t] < eps for all t < k-1, 2) for any k' \ge k, given any (k'+1)-wise constant function for which these moment bounds hold, one can run a vector-valued ODE to obtain a k'-wise constant function satisfying the same bounds, so by iterating this we get a k-wise constant function, and 3) because steps 1) and 2) hold for any eps>0, by a compactness argument there should exist a k-wise constant function satisfying these moment bounds for eps = 0. The argument for ReLU regression follows this same recipe except step 1) is more involved, because ensuring that the function they exhibit has nontrivial correlation with a ReLU is more complicated. UPDATE: Thanks for incorporating the comments! My opinion that this is a strong submission that should be accepted remains unchanged.

Strengths: This work settles a well-known open problem in agnostic learning that has been around since KKMS'08, and the proof is very impressive from a technical standpoint. The result should be of interest to any theorist working on distribution-specific supervised learning problems, and the tools it introduces for constructing moment-matching instances should be broadly useful to theorists interested in understanding the limitations of low-degree polynomials in learning.

Weaknesses: One might object that the question of hardness of getting OPT + eps over Gaussian marginals is purely of theoretical interest, because if the goal is simply to get O(OPT) + eps, poly-time algorithms exist for both problems, as the authors acknowledge. That said, given this is a purely theoretical work, I don't think this is a real weakness. One might also object that unlike the previous d^{log 1/eps} lower bounds, this only shows hardness for SQ algorithms as opposed to hardness for general algorithms under a standard assumption, but for the problems this paper considers, an SQ lower bound is compelling enough evidence for hardness, so I don't view this as a real weakness either.

Correctness: I have convinced myself the proofs are correct.

Clarity: The paper is clearly written. A minor comment: the argument in Lemma 3.8 is nice and seems quite non-standard to me among moment-matching constructions. Perhaps the authors can preface it with some high-level overview of the proof.

Relation to Prior Work: Yes, the prior work is explained clearly. For the camera-ready version, it would be good to mention how the bounds and techniques compare to that of https://arxiv.org/pdf/2006.15812.pdf (which appeared online after the deadline).

Reproducibility: Yes

Additional Feedback: - In the calculation at line 244, I think there should be a "-1" in the definition of chi-squared, and the Pr_{z'}[f(z') = 1] should be Pr_{z'}[f(z') = 1]^2 in the denominator so that the final expression is 4*1/2 - 1 = 1. - Relatedly, while chi_D(D_1,D_2) is defined, I think the definition for chi-squared divergence is missing. - Last paragraph of technical overview: "For the case *of* ReLUs" - The wording at the beginning of Section 3.1 of "establish the existence of a 1D Boolean function whose first k moments match the first k moments of a standard Gaussian" was initially confusing to me as you could always take the constant 1 function and E[z^t * f(z)] trivially agrees with the moments of z~N(0,1). But I guess the important point is to eventually get a nonzero function for which E[z^t * f(z)] = *0*. This is spelled out clearly in the technical overview, but maybe it would be good to reword the beginning of Section 3.1 to make this point abundantly clear.


Review 4

Summary and Contributions: The paper shows nearly optimal statistical query (SQ) lower bounds for agnostically learning halfspaces and RELUs under Gaussianly distributed data. Since agnostic weak learning of halfspaces is computational hard in the distribution-independent setting, a line of work has focused on agnostically learning halfspaces under natural distributions such as Gaussians. The best algorithm for this task runs in d^{O(1/eps^2)} time while the best known lower bound is a reduction from learning sparse parities with noise showing a runtime lower bound of d^{\Omega(log(1/eps))}. For the RELU variant, the best algorithm runs in time d^{poly(1/eps)} while a similar lower bound also follows from learning sparse parities with noise. This paper shows SQ lower bounds of d^{\Omega(1/eps)} and d^{poly(1/eps)} for these two problems, respectively, nearly matching existing algorithms. UPDATE: Thank you to the authors for the response and changes. My review remains that this paper should be a strong accept.

Strengths: Learning halfspaces is a fundamental and widely studied problem in learning theory. Because the general agnostic case is hard, learning halfspaces under Gaussian data is arguably the most natural interesting formulation. This paper’s main strength is that it makes an important step in pinning down the optimal runtime possible for this problem. As SQ captures all existing algorithms for learning halfspaces, SQ lower bounds seem to be pretty convincing for this problem. From a technical perspective, this is an interesting paper. The idea of following curves in the level sets of M(z) to reduce the number of nonzero entries in z in proof of Proposition 3.3 (through Lemma 3.8) is very clever and seems novel. Overall, the paper makes an important contribution, is technically interesting and I think should be accepted.

Weaknesses: Some of the proofs of the main SQ lower bounds are a somewhat routine application of the framework developed in DKS17. The hard instances for learning half spaces are the “hidden direction” Gaussian distributions (Definition 3.6) from DKS17 that resemble the “pancake” constructions for lower bounds for learning Gaussian Mixture Models in DKS17. However, I don’t think this is a serious drawback. It seems as though the construction of the hidden distributions A in this paper (e.g. using Proposition 3.3) required significant new ideas on top of DKS17.

Correctness: The proofs appear to be correct and are well-explained. This is a purely theoretical paper that shows lower bounds and thus has no empirical component.

Clarity: The paper is noticeably well-written and organized. This is a strength of the paper.

Relation to Prior Work: Section 1.1 does a very good job of discussing how this work fits into the surrounding literature.

Reproducibility: Yes

Additional Feedback: - On pg. 5, you say you prove Lemma 3.2 here but it is deferred to the appendix - k-decision lists are mentioned before being defined anywhere - I think there might be a missing -1 in the definition of chi^2 on line 244

[Author Response · NeurIPS 2020]

We thank the reviewers for their careful consideration of our paper and their uniformly positive feedback. We will
incorporate all minor comments and typos in the final version of our paper. Below we address specific questions and
comments by the reviewers.

**Reviewer 1**

Our paper establishes SQ lower bounds against weakly learning an unknown function in the given class. As correctly
pointed out by the reviewer, our lower bound construction produces instances in which OPT is close to $1/2$. Proving
SQ lower bounds for the case that OPT is a small constant is left as an interesting open question that may require
additional ideas.

The reviewer is correct that our lower bound works for the (broader) class of SQ algorithms – as opposed to only CSQ –
even for ReLUs, essentially because the class of hard functions are boolean-valued. We will clarify this point in the
final version.

We will add the definition of the unsupervised SQ dimension in our revised version, as well as the definition of
$\chi^2$-divergence.

We will add intuition regarding the notion of correlation. Intuitively, one can always think correlation as a metric of the
closeness of two functions. For boolean-valued functions, correlation is closely related to the probability the functions
disagree/agree. So, finding a function with a high correlation is the same as finding one with small error.

**Reviewer 2**

As pointed out by the reviewer and explained in the discussion section of our paper, our SQ lower bounds are
*qualitatively* optimal, up to a degree of the polynomial in the exponent. In particular, we prove an SQ lower bound of
$d^{\Omega(1/\epsilon)}$ for agnostically learning LTFs (under Gaussian marginals). In comparison, the best known algorithm for the
problem has runtime $d^{O(1/\epsilon^2)}$; and the best previous lower bound was $d^{\Omega(\log 1/\epsilon)}$.

It remains an interesting open question for future work whether $d^{\Omega(1/\epsilon^2)}$ is an SQ lower bound for the problem.

Typos/Definition: We will revise line 42 and add the definition of a $k$-decision list in the final version.

Regarding the potential existence of faster PTAS for the problem: Suppose there was an algorithm for agnostically
learning LTFs under Gaussian marginals that achieves error $(1 + \gamma)\text{OPT} + \epsilon$ and runs in time $\text{poly}(d^{\log(1/\gamma)}, 1/\epsilon)$.
Then, by setting $\gamma = \epsilon$ and using the fact that OPT is at most 1, we would obtain an algorithm with error $\text{OPT} + 2\epsilon$
that runs in quasi-polynomial time (as a function of $1/\epsilon$).

Proof of Proposition 3.3: We will add a clarification to line 274. In the proof, we need the breakpoints to be distinct, (as
is stated in the beginning of the proof). In the proof of Proposition 3.3., we work with functions that have at most $k + 1$
breakpoints, which means that for some $\epsilon$ it is possible that for some values of $i$, $b_i$ and $b_{i+1}$ may be equal. Of course,
the compactness argument shows the existence of such a function and indeed it may have less than $k + 1$ breakpoints,
which will in fact yield a function with higher correlation. (Note for example that, if we knew that the function has $\sqrt{k}$
breakpoints, we could improve the lower bound to $d^{\Omega(1/\epsilon^2)}$.)

**Reviewer 3**

Thank you for pointing out this concurrent related work. We will add a paragraph with a comparison to our results and
techniques in the revised version of our paper.

Thank you for the detailed technical comments. We will address them in the final version. We will also add prose
providing the intuition behind Lemma 3.8.

**Reviewer 4**

Thank you for pointing out these typos. We will fix them in the final version and add the definition of $\chi^2$-divergence.

[Meta-Review · NeurIPS 2020]

The paper studies a fundamental problem and provides new lower bounds that come close to matching know upper bounds. The reviewers all strongly support accepting the paper. They also find the techniques used in the paper interesting in their own right. Added after decisions: During the review process, we noticed that the problem and results in this submission are closely related to those in another submission. After I had written my original metareview, it was further brought to my attention that both of these submissions are on arXiv (the other one is arXiv:2006.15812) and the arXiv versions already acknowledge the parallel work. Now that both papers have been accepted to NeurIPS, I'm asking both sets of authors to include in their final versions a discussion explicitly comparing their results to those in the other paper.